# Enhancers and genome conformation provide complex transcriptional control of a herpesviral gene

David W Morgens [1 ✉], Leah Gulyas [1], Xiaowen Mao[1], Alejandro Rivera-Madera [2], Annabelle S Souza[2] & Britt A Glaunsinger [1,2,3 ✉]

## Abstract

**Complex transcriptional control is a conserved feature of both eukaryotes and the viruses that infect them. Despite viral genomes being smaller and more gene dense than their hosts, we generally lack a sense of scope for the features governing the transcriptional output of individual viral genes. Even having a seemingly simple expression pattern does not imply that a gene's underlying regulation is straightforward. Here, we illustrate this by combining high-density functional genomics, expression profiling, and viral-specific chromosome conformation capture to define with unprecedented detail the transcriptional regulation of a single gene from Kaposi's sarcoma-associated herpesvirus (KSHV). We used as our model KSHV ORF68 – which has simple, early expression kinetics and is essential for viral genome packaging. We first identified seven cis-regulatory regions involved in ORF68 expression by densely tiling the ~154 kb KSHV genome with dCas9 fused to a transcriptional repressor domain (CRISPRi). A parallel Cas9 nuclease screen indicated that three of these regions act as promoters of genes that regulate ORF68. RNA expression profiling demonstrated that three more of these regions act by either repressing or enhancing other distal viral genes involved in ORF68 transcriptional regulation. Finally, we tracked how the 3D structure of the viral genome changes during its lifecycle, revealing that these enhancing regulatory elements are physically closer to their targets when active, and that disrupting some elements caused large-scale changes to the 3D genome. These data enable us to construct a complete model revealing that the mechanistic diversity of this essential regulatory circuit matches that of human genes.**

**Keywords** Capture Hi-C; CRISPR Interference; Gene Regulation; Herpesvirus; KSHV
**Subject Categories** Chromatin, Transcription & Genomics; Microbiology, Virology & Host Pathogen Interaction

## Introduction

Underlying the lifecycle of all DNA viruses is a highly regulated cascade of viral gene transcription. In human herpesviruses, many of these genes are transcribed in a host-like manner; nuclear dsDNA viral genomes can be chromatinized by human histones (Kutluay and Triezenberg, 2009), decorated with enhancer marks (Toth et al, 2010), driven by human transcription factor binding (Qi et al, 2019), and even form transcription-associated domains with human CTCF and cohesion (Stedman et al, 2008; Campbell et al, 2018). Exhaustively identifying and characterizing these regulatory features can both help us understand the biology of human pathogens such as Kaposi sarcoma-associated herpesvirus (KHSV)—a major cause of cancer in AIDS and other immuno-compromised patients—and also contribute to our knowledge of human transcriptional regulation.

Numerous examples of noncoding regulatory sequences have been found in DNA viruses. Enhancer sequences include the first described enhancer on SV40 (Banerji et al, 1981) the EIIA enhancer on adenovirus (Loeken and Brady, 1989), the major immediate enhancer element in the betaherpesvirus human cytomegalovirus (Stinski and Isomura, 2008; Dooley and O'Connor, 2020), and a recent proposal that the terminal repeats of KSHV act as enhancers (Izumiya et al, 2024). Other viral regulatory elements act through the expression of noncoding elements. In murine gammaherpesvirus 68, there are tRNA-like elements that control latency and egress (Feldman et al, 2016; Hoffman et al, 2019). In KSHV, many different functional elements beyond coding mRNAs are transcribed, including miRNAs that regulate cancer phenotypes (Hu et al, 2015; Gay et al, 2021), origin RNAs that regulate viral DNA replication (Wang et al, 2006, 2004), circular RNAs (Tagawa et al, 2018, 2021), and long ncRNAs with various functions (Rossetto and Pari, 2011; Campbell and Izumiya, 2020; Sun et al, 1996; Schifano et al, 2017; Chandriani et al, 2010). While these individual events have been explored to different degrees, systematic searches for regulatory sequences have been limited by traditional methods that rely on deletions to perturb noncoding elements, which in the densely encoded KSHV genome may have unintended effects on surrounding elements.

For the human genome, the discovery of functional regulatory sequences has been greatly accelerated by the use of CRISPR interference or CRISPRi (Hilton et al, 2015; Thakore et al, 2015; Liu

[1]Department of Plant and Microbial Biology, UC Berkeley, Berkeley, CA, USA. [2]Department of Molecular and Cell Biology, UC Berkeley, Berkeley, CA, USA. [3]Howard Hughes Medical Institute, UC Berkeley, Berkeley, CA, USA. ✉E-mail: dmorgens@berkeley.edu; glaunsinger@berkeley.edu

et al, 2017; Fulco et al, 2016; Tycko et al, 2019). Notably, by recruiting repressive chromatin regulators to DNA, CRISPRi can repress gene expression not only through proximal promoter elements but also by perturbing distal, enhancer elements. While this tool has been applied widely on the host, it can also effectively repress transcription from the KSHV genome (Brackett et al, 2021), and thus has the potential to provide deep insight into the components and structure of viral transcriptional networks. While searches for human regulatory regions are limited to predicted enhancers or nearby regions, the viral regulatory regions will be necessarily contained within the relatively compact viral genome and thus be amenable to exhaustive characterization.

Here we combine CRISPRi with a library of guide RNAs densely tiling the KSHV genome, allowing a thorough interrogation of potential regulatory activity controlling the expression of a single viral gene, ORF68. ORF68 was selected as a proof of concept for this study as it has a simple early-expression pattern (Gabaev et al, 2020; Arias et al, 2014) but plays an essential role late in the viral life cycle during the packaging of new viral DNA (Gardner and Glaunsinger, 2018; Didychuk et al, 2021). By complementing CRISPRi with a Cas9 nuclease screen and transcriptional profiling, we identified promoters that control expression through their associated coding regions, as well as noncoding regulatory elements that comprise a surprisingly sophisticated network to regulate ORF68 transcription. Finally, we use viral-specific chromosome conformation capture to measure physical interactions between regulatory regions and their distal targets and demonstrate how disruption of these regions changed the 3D structure of the viral genome. These findings illustrate the power of this approach for mapping viral gene regulatory networks on a genome-wide scale.

## Results

### CRISPRi tiling identifies regulatory regions across the viral genome

We sought to define the layers of regulation underlying the expression of an individual viral gene, using the nuclear replicating dsDNA virus KSHV. We selected KSHV ORF68 as our model gene, as it is required for progeny virion production, has no known direct transcriptional regulators, and its expression initiates early in the lytic cycle and stays on throughout the rest of the viral lifecycle (Gabaev et al, 2020; Arias et al, 2014). KSHV gene regulation can be readily studied using the renal carcinoma cell line iSLK, a well-established model for the KSHV lifecycle that includes a doxycycline-inducible version of the KSHV lytic transactivator ORF50 to facilitate lytic reactivation from latency.

We began the construction of this regulatory network by querying how the silencing of each KSHV locus using CRISPRi influenced ORF68 expression. We latently infected iSLK cells with a version of KSHV containing a HaloTag fused to the N-terminus of ORF68 at the endogenous locus (HaloTag-ORF68; (Morgens et al, 2022), allowing us to directly measure ORF68 protein levels, as well as a constitutively expressed GFP reporter that marks infected cells. Additionally, we lentivirally introduced a constitutive dCas9-KRAB fusion (CRISPRi), which would be recruited to a targeted viral region upon delivery of a sgRNA. We then delivered a library of guide RNAs densely tiling the KSHV genome with an average of

one guide every eight basepairs (Morgens et al, 2022). After four days, the virus was reactivated, and cells were treated with a fluorescent HaloTag ligand to monitor ORF68 protein levels. Twenty-four hours post-reactivation, cells were fixed and sorted for high and low ORF68 protein expression (Fig. 1A). By sequencing the sgRNA locus from both populations, we calculated an average guide enrichment from two replicates. A negative value signifies that the guide was enriched in the low ORF68 signal population, indicating that silencing that locus inhibits the expression of ORF68. Similarly, a positive value signifies that the guide was enriched in the high ORF68 signal population, indicating that silencing that locus increases the expression of ORF68 (Fig. EV1A,B).

We performed a sliding window analysis and identified 8 loci with target guide RNA scores that differed significantly from negative controls (Fig. 1B; Dataset EV1, 2). Our strongest signal corresponded to a peak poised immediately upstream of HaloTag-ORF68 itself, confirming successful transcriptional inhibition of the locus by CRISPRi. We observed that the center of most other peaks also corresponded with known transcriptional start sites (TSSs); since we expect CRISPRi to work primarily by impeding transcription, these peaks are named by their nearest TSS for simplicity (Fig. EV1C–H). For example, we find one peak near the TSS of ALT, a lncRNA of unknown function which runs antisense to many genes expressed during latency (Schifano et al, 2017), that we will refer to as TSSalt (Fig. EV1E). The exceptions include one peak that roughly maps to the ORF50 coding locus (Fig. EV1C)— likely targeting the exogenous copy of ORF50 used to reactivate the virus—as well as an additional peak which does not correspond to a previously described TSS (Fig. EV1H) (Ye et al, 2019) but is located near the shared polyA sites of ORF75, ORF74, K14, K15, and the ALT lncRNA (Schifano et al, 2017; Gregory Bruce et al, 2017). We will refer to these as ORF50 and polyA75, respectively.

We next validated these eight regions' effect on ORF68 by delivering a pool of three CRISPRi guides per locus and measuring the fraction of cells expressing HaloTag-ORF68 at two timepoints post-reactivation (Fig. 1C). These confirmed our screen results, but we did note that some of the effects on ORF68 levels were not sustained at later timepoints, most notably when targeting TSSalt. This may reflect changing regulatory events as the viral lifecycle progresses. While guides targeting the EF1a promoter displayed the expected effect as well, these guides were highly toxic, most likely through their association with the BAC-encoded EF1a-EGFP-HygroR selection locus or the possible inhibition of the host EF1a site. Thus, we excluded this peak from further analysis.

Despite the correspondence of most peaks to TSSs, the large observed footprint of CRISPRi prevented the high-resolution identification of important underlying regulatory features. Illustrating this, the TSS68 peak is comprised of guides targeting not only the entirety of the ORF68 coding region, but also many surrounding genes (Fig. 1D). This equates to approximately a 2–5 kb window of CRISPRi repression from a single guide, which is mirrored at other loci (Fig. EV1D). This large footprint is likely due to the spread of KRAB-induced heterochromatinization (Lensch et al, 2022) and should be taken into account when using sodium butyrate and CRISPRi on viral genomes, which display significantly higher gene density than cellular genomes. This prompted us to further interrogate the global and local transcriptional effects of CRISPRi at each regulatory region on the viral genome.

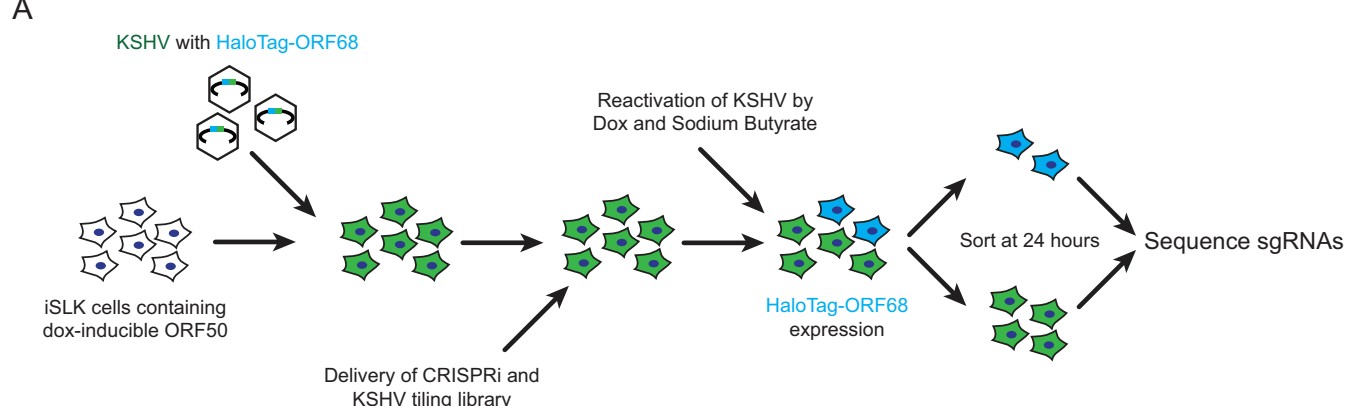

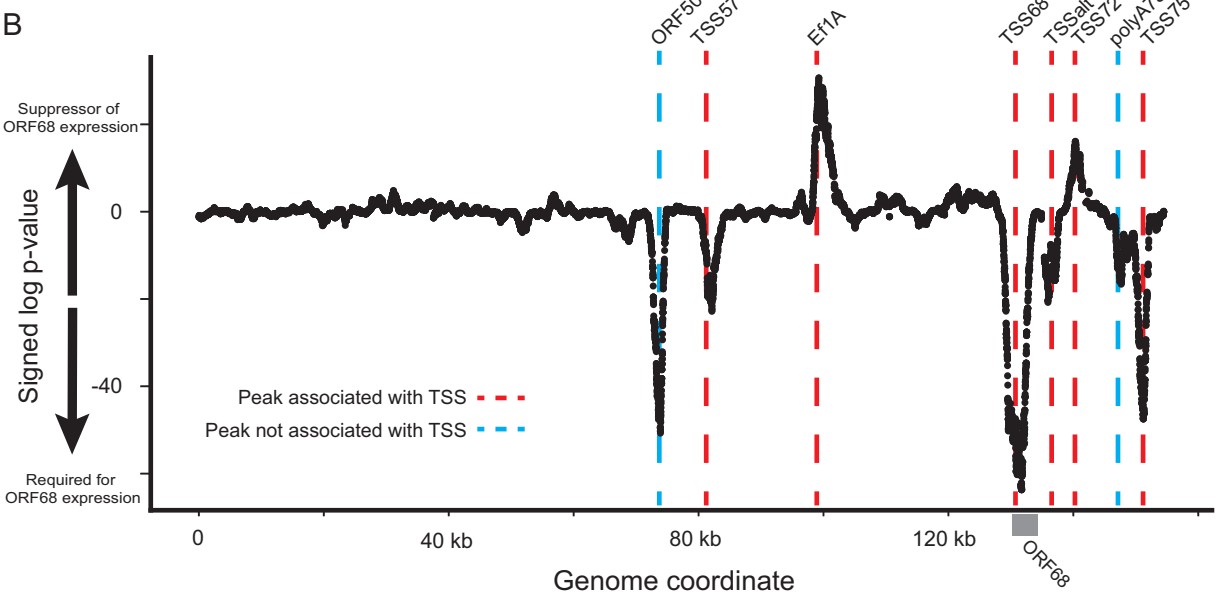

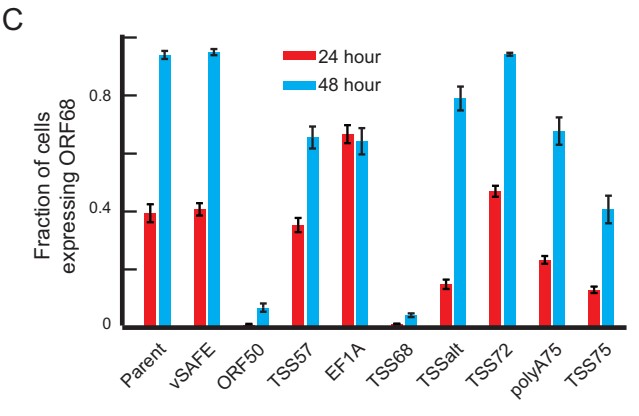

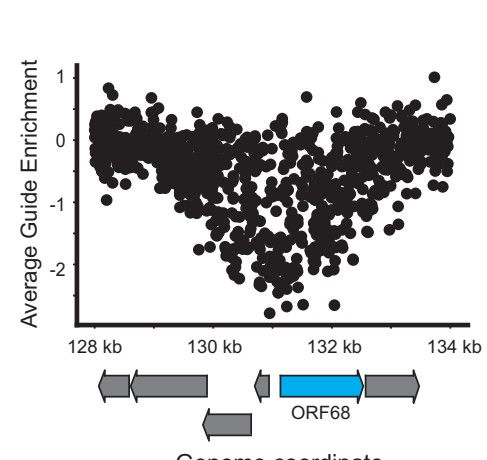

**Figure 1. CRISPRi screen identifies novel viral regulatory regions.**

(A) Schematic of screen. The viral genome encodes a constitutive fluorescent marker (green) and a HaloTag-ORF68 fusion (blue). (B) Summary of results from the CRISPRi screen. The X-axis identifies the genome coordinate on the BAC16 KSHV genome. The Y-axis represents the log-transformed *p* value of each locus relative to the negative control distribution. Red dotted lines indicate the location of transcriptional start sites. Blue dotted line identifies the two peaks that do not associate with a TSS. (C) Validation of pooled guides targeting each peak. Three guides were used to target each locus identified on the x-axis. The Y-axis displays the mean fraction of cells expressing the HaloTag-ORF68 for the 24 or 48 h post-lytic reactivation timepoints. Error bars represent the standard error of the mean from four replicates from independent reactivations. (D) Enrichment of individual guides at the ORF68 locus. Each dot represents a single guide, with the target location displayed on the x-axis and the average enrichment from two replicates on the y-axis. Arrows represent coding regions of ORF68 (in blue) and surrounding genes in gray. Source data are available online for this figure.

## CRISPRi recruitment to the viral genome inhibits many genes locally

The regulatory regions identified above could either impact ORF68 transcription specifically or could more globally disrupt KSHV lytic gene expression. To evaluate these possibilities, we performed polyA+ RNA-seq at 24 h post-reactivation on the previously validated guide pools (Fig. EV2A). Silencing of the regulatory loci by CRISPRi caused changes in ORF68 mRNA levels that are consistent with those observed at the protein level by both RNA-seq and follow-up RT-qPCR (Figs. 2A and EV2B,C; Dataset EV3), suggesting that regulation primarily controls RNA abundance. Nearly all viral genes were downregulated when ORF50, TSS75, or TSS57 were silenced, aligning with the critical roles of ORF50, ORF75, and ORF57 proteins in the viral lifecycle. ORF50 is a master regulator of KSHV lytic reactivation (Guito and Lukac, 2012), ORF75 likely acts indirectly to prevent innate immune suppression of viral gene expression (Full et al, 2014; McCollum et al, 2023), and ORF57 has many reported viral functions including the export of viral mRNA from the nucleus (Majerciak and Zheng, 2015).

In contrast, guides targeting TSS68 and TSSalt had a limited global effect, strongly inhibiting only a small number of genes each. The strongest downregulated genes fall within the local region of the guides (Fig. 2B,C), reaffirming that CRISPRi inhibits transcription in a region of ~2.5 kb around the targeting site. This appears true even when the effect of CRISPRi increases ORF68 mRNA expression, as in the case of TSS72. Silencing the TSS72 region causes a global increase in viral gene expression (Fig. 2A) despite still locally decreasing transcription (Fig. 2D). Many of these locally repressed regions include ORF73, which encodes for LANA, whose expression is required for latency maintenance (Qi et al, 2019; Uppal et al, 2014), though whose knockdown was not seen here to affect ORF68 expression. To test whether targeting TSS72 is affecting latency, we grew our knockdown lines in the absence of hygromycin selection and observed a more rapid loss of latency only when targeting TSS75 or polyA75 (Fig. EV2D). We also tested these guides in a naturally infected, CRISPRi+ B-cell line—TREx-RTA-BCBL1 (Brackett et al, 2021; Nakamura et al, 2003)—and found only guides targeting ORF50 and TSS68 had large effects on ORF68 transcription (Figure EV2E). Thus, RNA-seq corroborates our screening data, and reveals both local and global effects of CRISPRi at our target loci.

We next tested whether these altered ORF68 and ORF75 RNA levels stem from transcriptional changes after CRISPRi targeting of the regulatory loci. At 24 h post reactivation, we used ethynyl uridine (EU) to metabolically label newly transcribed RNA for two hours. EU-labeled RNA was purified, modified with biotin, and

isolated, and the levels of newly synthesized ORF68 and ORF75 mRNA were measured by RT-qPCR (Fig. 2E). As expected, targeting TSS68 or TSS75 had a dramatic effect on their respective nascent RNAs, and silencing of other regulatory loci yielded a reduction or increase in EU-labeled RNA consistent with total RNA levels. The exception is TSS57, whose silencing did not significantly reduce nascent ORF68 mRNA at 24 h post reactivation. Thus, while these regulatory loci all impact the total RNA abundance of their targets, they may do so through a combination of transcriptional and post-transcriptional mechanisms.

## Viral knockouts identify associated coding features

While CRISPRi at a given locus may effectively suppress multiple viral genes, not every gene will be responsible for the observed effect on ORF68 expression. For example, while guides targeting the TSS68 repress expression of ORFs 65, 66, 67, 68, and 69 (Fig. 2B), downregulation of the ORF68 reporter is most likely due to direct repression of the ORF68 promoter. Furthermore, CRISPRi alone is unable to distinguish whether regulatory regions influence the transcription of a regulatory protein or act by a noncoding mechanism. Therefore, we next directly assessed the role of coding loci underlying each regulatory locus by performing a CRISPR nuclease tiling screen, using a version of the HaloTag-ORF68 reporter iSLK line containing Cas9 instead of CRISPRi.

In previous screens, we have reported a strong background effect, where targeting any locus on the viral genome with Cas9 nuclease resulted in a decrease in reporter expression (Morgens et al, 2022). Here, we observed unexpectedly that this background effect was variable across the viral genome, with targeting the region upstream of the ORF68 coding region having a stronger effect on reporter expression than targeting downstream (Fig. 3A; Dataset EV4,5). The reason for this difference is unknown (possibly a local effect of DNA damage on the viral genome), but to adjust for the differences in the local background, we used a boundary method to identify coding regions: for each candidate coding exon, we compared the enrichment of the coding region to the immediate adjacent noncoding region, and considered the coding region a hit if the boundaries were both significantly increased, or both significantly decreased.

This yielded an exhaustive list of viral coding regions that control ORF68 expression at 24 h post-reactivation: ORF50, HygroR, ORF68, and ORF75 (Fig. 3B–E). Targeting the corresponding CRISPRi loci of ORF50, TSS68, and TSS75 each disrupts ORF68 expression, while CRISPRi repression of the EF1a promoter controlling HygroR expression increased ORF68 expression. Loss of functional protein thus likely explains these regions' effect on

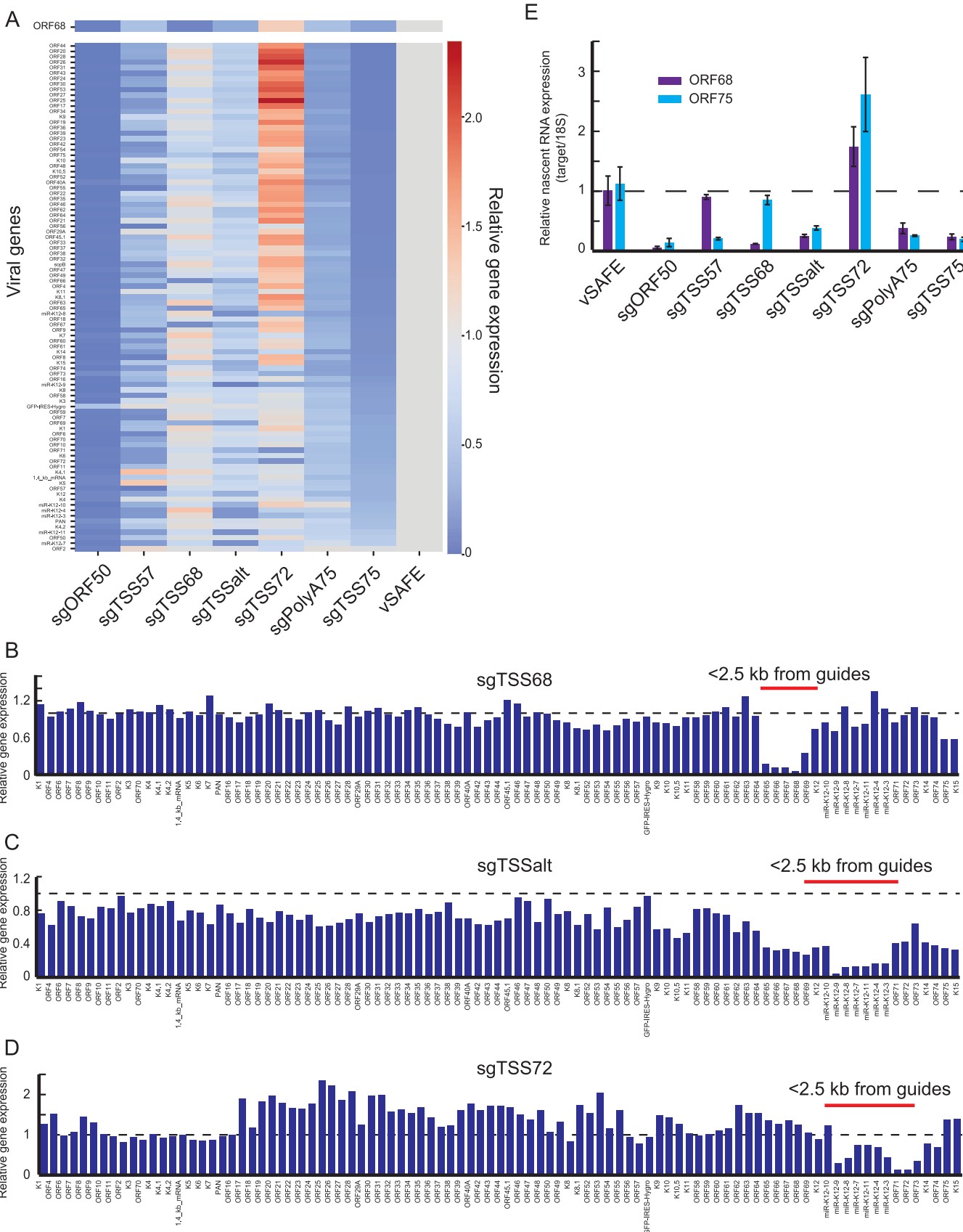

**Figure 2. Local effects caused by CRISPRi.**

(A) Heatmap of changes to viral gene expression at 24 h relative to vSAFE negative controls, with ORF68 presented at the top. Each column shows relative gene expression changes following CRISPRi-induced suppression of the indicated locus. Values are sorted by the effect of sgTSS75. Average of three replicates. (B–D) Change in RNA level of each viral gene relative to vSAFE negative controls in genome order. Genes whose start codons are within 2.5 kb of at least one guide in the targeting pool are highlighted. (E) Nascent RNA expression was measured by RT-qPCR at 24 h post-reactivation from cells treated for 2 h with EU. Mean values are presented relative to parental cells, and error bars are standard errors from three independent replicates. Source data are available online for this figure.

ORF68. In contrast, we observe no effect on ORF68 by targeting ORF57 and ORF72 coding regions with Cas9 (Fig. 3F,G). Thus, TSS57 and TSS72 presumably impact ORF68 independently of their associated coding ORFs. The final two CRISPRi loci, TSSalt and PolyA75, are not located at the promoters of any coding regions, and indeed we find no evidence of a coding region near TSSalt that affected ORF68 expression. PolyA75 is adjacent to the ORF75 coding locus, but no other coding elements are identified that could explain its activity. These regions with no associated coding region thus likely act through noncoding elements that are unable to be efficiently disrupted by Cas9.

To test whether the identified coding regions have the expected negative effect on the mRNA levels of ORF68, we cloned pooled guides for nuclease-targeting at the coding regions of ORF50 and ORF75, including ORF57 as a negative control. ORF50 and ORF75 pools had the expected effect of decreasing ORF68 protein levels (Fig. 3H) as well as corresponding depletion of ORF68 RNA (Fig. 3I). To determine whether ORF50 and ORF75 are direct or indirect activators of ORF68, we transfected a reporter plasmid containing 230 bp of upstream of the ORF68 start codon driving a HaloTag reporter along with an expression plasmid of either ORF50 or ORF75 into HEK293T cells. Only ORF75 caused a significant increase in expression from the ORF68 promoter (Fig. 3J). Given that ORF50 is required for ORF75 expression (Fig. 3I), these data suggest a regulatory model of protein-coding elements where ORF50 activates ORF75, which, in turn, activates ORF68. We also observed that targeting either ORF50 or ORF75 reduced RNA levels of ORF57 (Fig. 3I). Interestingly, sgORF57 pools reciprocally decreased ORF75 RNA levels, yet ORF57 protein disruption had little or no effect on ORF68 mRNA levels—as predicted by the nuclease screen—suggesting an unclear regulatory relationship.

## Mapping regulatory targets of noncoding elements

The remaining CRISPRi peaks at TSS57, TSSalt, TSS72, and polyA75 presumably are noncoding loci that instead act in a distal manner to indirectly impact one of the three viral proteins that affect ORF68 expression. We therefore returned to our RNA-seq data and measured the correlation between each sample (Figs. 4A and EV3A). We hypothesized that despite the local effects of CRISPRi, the regulatory regions would correlate most strongly with their regulatory target, and as we have identified all regulatory regions, the number of potential targets is limited. TSSalt and polyA75 most strongly correlated with TSS68 and TSS75, respectively, along with increasing their expression (Fig. 2E), suggesting that these regions promote ORF68 and ORF75 expression. Conversely, TSS72 had a strong anticorrelation with polyA75 and TSS75. Given that recruitment of CRISPRi to TSS72 causes an increase in viral gene expression, including increased transcription of ORF75, this suggests that TSS72 acts to repress the expression of ORF75. TSS57 weakly correlated with many targets, preventing any firm conclusion. We can thus use these correlative regulatory interactions along with data

from Fig. 3 to create a model regulatory network consisting of both coding and noncoding elements controlling ORF68 expression (Fig. 4B). Of note, as we performed these experiments while overexpressing ORF50 from an exogenous promoter, we were likely unable to detect any regulation of the viral copy of ORF50. While silencing TSS57, ORF57, or TSSalt each decreased transcription of ORF75 (Figs. 2E and EV2C, 3I), we have excluded these interactions from our model as we did not observe the expected subsequent changes to global or ORF68 transcription (Fig. 2A).

To test this model, we evaluated how the components of this ORF68 regulatory network impacted virion production in KSHV-infected cells using a supernatant transfer assay. BAC16-derived KSHV expresses GFP, which enables the quantification of infected recipient HEK293T cells by flow cytometry. As expected, targeting any of the coding elements via their promoters (TSS57, TSS68, and TSS75) caused a severe loss in infectious virion production (Fig. 4C). Targeting polyA75 also reduced virion production, albeit more modestly, consistent with the regulatory network. In contrast, guides targeting TSS72 or TSSalt did not negatively impact virion production (Fig. 4C). This was expected for TSS72, whose silencing increases ORF68 expression. However, that targeting TSSalt—which specifically disrupts ORF68 expression at the early 24 h but not the late 48 h late time point (see Fig. 1C)—did not impair virion production suggests that while ORF68 expression is essential late in infection, it may be dispensable at early timepoints. We also targeted these regions in CRISPRi+ BCBL1 cells, and found that ORF57, ORF68, and ORF75 expression were all essential for the production of infection viral particles along with the polyA75 site (Fig. EV3B).

Previous work has demonstrated the use of CUT&RUN to evaluate protein binding and histone modifications on the KHSV genome (Ye et al, 2024; Kumar et al, 2022). To evaluate the chromatin marks present at each regulatory region, we performed CUT&RUN against H3K27ac and H3K4me1 in wildtype, sgORF50, and sgORF75 knockout lines (Figs. 3D and EV3C; Dataset EV8). Upon reactivation, we observed a loss of global histone signal, which was not dependent on ORF50 expression—likely instead influenced by the presence of the histone deacetylase inhibitor sodium butyrate. During reactivation, the noncoding regulatory regions TSSalt and TSS72 are within a large peak of H3K27 acetylation—consistent with enhancer activity—though were devoid of the poised enhancer mark H3K4 monomethylation. In contrast, the polyA75 locus was H3K4me1 positive and H3K27ac negative, consistent with its lack of an associated transcriptional start site. While this enhancer mark was generally absent from the promoter regions of ORF50, ORF57, and ORF68, we did note its presence at TSS75 despite this region acting at least partially through its promoter activity and associated ORF75 protein. We also used CUT&RUN to examine CTCF binding to the viral genome and found it was highly stable in each condition—with one of the only dynamic CTCF peaks appearing during reactivation at

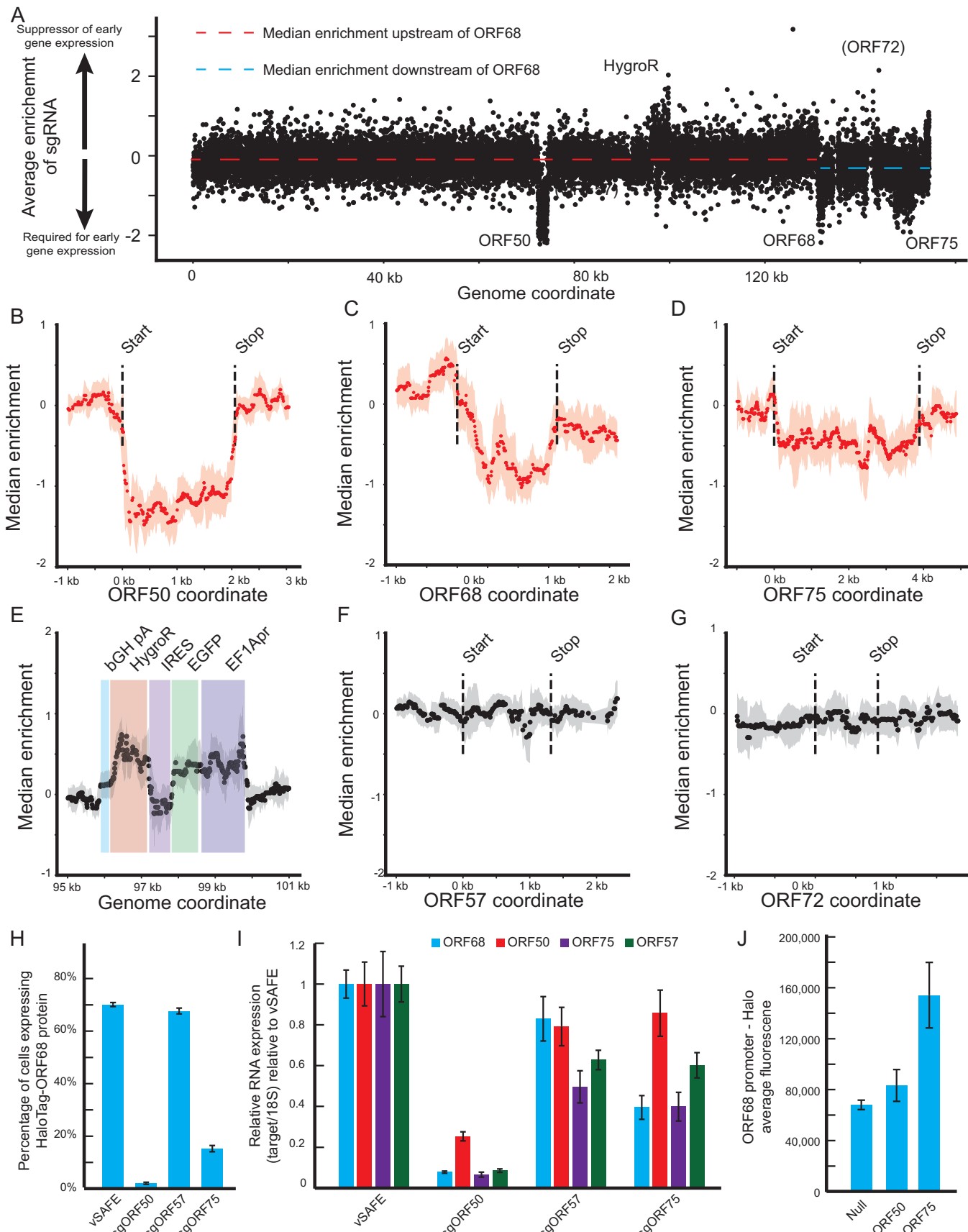

◀ **Figure 3.   Knockout screen maps associated coding regions.**

(A) Enrichment of individual guides. Each dot represents a single guide, with the target location displayed on the x-axis and the average enrichment from two replicates on the y-axis. Red and blue dotted lines represent the median guide enrichment for the two regions indicated. (B-G) Median smoothed enrichments from Cas9 nuclease screen at associated coding locus. Dotted lines indicate exon boundaries. For each guide, the median enrichment of a 500 bp window centered at the target locus was calculated along with an IQR. The median value is shown as a point, and IQR is shown as a shaded region. Regions were considered significant (shown in red) if the guides on each side of the boundary were significantly different and in consistent directions. For the EGFP-HygroR locus, the location of functional units is shown in color. (H) Percent of cells expressing HaloTag-ORF68 24 h post-reactivation for the indicated pool of coding region-targeting guides. Values are averages of four independent replicates, and error bars represent standard error. (I) RT-qPCR measurements of ORF68, ORF50, ORF75, and ORF57 mRNA at 24 h post-reactivation following Cas9-based targeting of the loci indicated on the x-axis. Mean data were presented relative to 18S RNA and vSAFE. Error bars show the standard error of the mean from four technical replicates. (J) Average fluorescence from HEK293T cells transfected with an ORF68 promoter-driven HaloTag and a plasmid expressing the indicated viral protein. Error bars are standard errors centered on the mean from seven independent replicates. Source data are available online for this figure.

the TSS72 locus (Figs. 3D and EV3C; Dataset EV8). While we observed few changes in CTCF binding, we next investigated how the 3D structure of the genome changed during this cycle and what the relative physical orientation of the regulatory elements was.

## Physical interactions of noncoding regions and their targets

We hypothesized that these noncoding elements at TSSalt, TSS72, and polyA75 function distally as either enhancers or regulators of the 3D viral genome structure. Enhancers are expected to be physically close to their target promoters, which can be determined by measuring genome architecture using chromosome capture techniques like Hi-C. Previous work has demonstrated that high-quality viral-viral or viral-host contacts can be obtained using capture Hi-C (Campbell et al, 2022; Kumar et al, 2022; Campbell et al, 2018). We thus evaluated this approach on iSLK-BAC16 cells 24 h post reactivation (Fig. 5A; Dataset EV7). Indeed, by applying KSHV-specific sequence capture to our libraries, we were able to achieve high (<2 kb) resolution contact frequency maps on the viral genome (Fig. EV4A) with relatively few sequencing reads (~7 million reads). Contact features were clear at 1–2 kb resolution but still notable at even higher (500 bp) resolution (Fig. EV4A).

We first asked whether the regulatory elements that interact functionally are also physically proximal, which would be consistent with their activity as enhancers. To function as an enhancer, TSSalt should have a high contact frequency with its target TSS68, while TSS72 and polyA75 should have a high contact frequency with their target TSS75. Indeed, this is the case: when looking at the contact frequency of these regions with the rest of the viral genome at 1 kb resolution, TSSalt, TSS72, and polyA75 are physically close to their identified targets but not to another regulator region, TSS57 (Fig. 5B,C). This also holds true in the reciprocal interactions (Fig. EV4B–D). These results are consistent with TSSalt, TSS72, and polyA75 functioning as enhancers and exclude the possibility that TSS57 itself is acting as an enhancer targeting either TSS68 or TSS75.

We next asked whether this physical proximity is greater than would be expected given the linear genomic distance between the elements and their regulatory targets. Previous Hi-C data has consistently identified that contact frequency is reduced with distance and that this decay can be modeled using a power law (Lieberman-Aiden et al, 2009; Zhou et al, 2020). However, the relationship between linear distance and the observed contact frequency on the viral genome diverged at higher distances (Fig. EV4E), presumably reflecting that the KSHV genome is present in circular and concatenated forms

(Campbell et al, 2022). Indeed, a distance metric adjusted for these forms better matched the power law expectation (Fig. EV4F) and was used for subsequent analyses.

By normalizing the measured contact frequencies (Fig. EV4A) to the expected contact frequency based on distance, we obtained a relative contact frequency that represents the structure of the viral genome at 1 kb resolution (Fig. 5D). In agreement with the close contacts shown in Fig. 5B,C, we observed a strongly associated region from ~129–154 kb that contains TSS68, TSSalt, TSS72, polyA75, and TSS75 (Fig. 5D, outlined region 1), indicating that these regulatory elements are physically closer than expected given their linear distance. Of note, this corresponds to the region with increased background in the Cas9 screen (Fig. 3A), consistent with disruptions to this region causing changes to local transcription. Compartments like this exist on the KSHV genome at a smaller scale than typical transcriptional-associated domains (TADs) (Campbell et al, 2022), so we instead refer to them as insulated regions. Other stand-out physical associations include the 129–154 kb insulated region and the 1–3 kb region containing promoters for K1, ORF4, and ORF6 (Fig. 5D, region 2), as well as the region around TSSalt (which includes the right lytic origin of replication) and the region at ~24 kb which includes the left lytic origin of replication (Fig. 5D, region 3). These physical associations may represent chromosomal loops between distal regions of the viral genome. However, evaluating whether these loops are statistically significant or if there are additional loops present would require a robust statistic, and it is unclear what would be appropriate given the small size of the viral genome, the high frequency of distal interactions, and the rectangular shape of the observed loops (Fig. 5D, regions 2 and 3).

## Changing physical relationship between regulatory regions

If ongoing transcription at these regulatory elements is important for physical interaction with their targets, we reasoned that silencing them could alter the architecture of the viral genome. We tested this by targeting each regulatory element using CRISPRi, then performing capture Hi-C to measure how the 3D structure of the viral genome changed. We modeled how the genome structure changes by grouping the resulting capture Hi-C maps into initial, intermediate, and final stages, representing progression through the viral lytic lifecycle and the regulatory network controlling ORF68 expression by introducing a roadblock caused by silencing each element (Fig. 6A; Dataset EV7). Maps from unreactivated cells and reactivated cells with silenced ORF50 (sgORF50) represented the initial stage, as the viral lytic cycle cannot progress in the absence of ORF50. The intermediate stage comprised the maps from

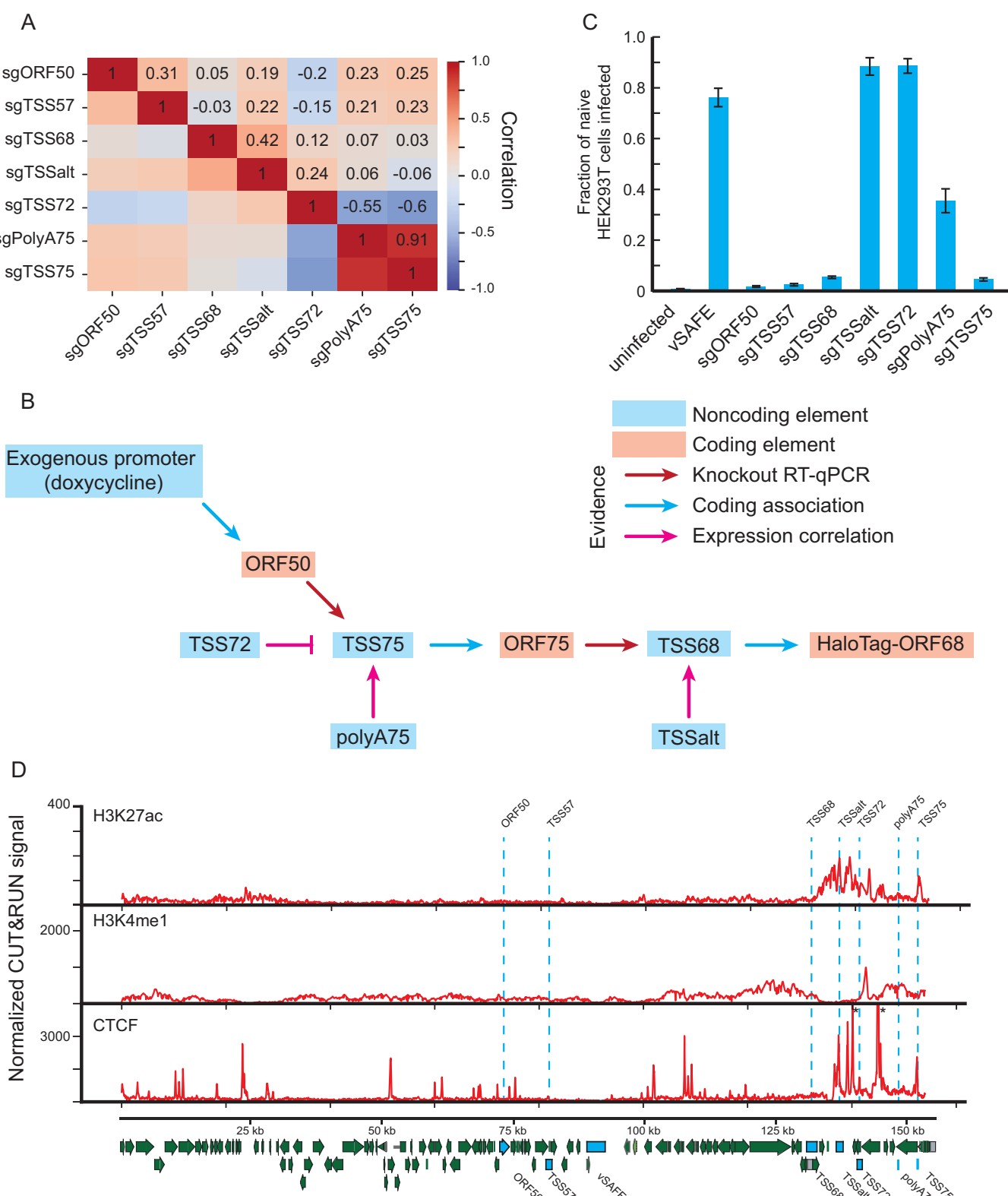

**Figure 4. Mapping regulatory network by effect on viral transcription.**

(A) Co-correlation matrix of the RNA-seq data from Fig. 2a. For each indicated pair of sgRNA pools, a Pearson correlation was calculated between the viral RNA levels. (B) Model of regulatory events controlling transcription of the ORF68 locus. (C) Supernatant transfer assay measuring changes in KSHV virion production after knockdown of the indicated loci. Error bars represent standard error centered on the mean from six independent reactivations. (D) CUT&RUN signal from indicated mark in reactivated iSLK cells 24 h post-reactivation. Asterisks indicate peaks above the maximum signal graphed. The signal is averaged from three independent replicates. Source data are available online for this figure.

reactivated cells with silenced TSS75, TSS57, and PolyA75, as the RNA-seq data indicated these cause significant changes to the viral transcriptome. The final stage comprised the maps from cells with silenced TSS68, TSSalt, the negative control vSAFE, and the negative regulatory element TSS72.

We first examined the insulated region at 129–154 kb, which contains many of these regulatory elements. At 1 kb resolution, we observed large structural changes to this region as the viral life cycle progresses from the initial (Fig. 6B) to the intermediate (Fig. 6C) to the final stage (Fig. 6D), with this 129–154 kb compartment only clearly forming in the final stage. We quantitatively assessed these structures using an insulator score, which was calculated from the relative frequency of contacts across a given region. Insulators "protect" expressing genes from the neighboring genomic environment; they were thus detected here as areas with fewer than expected contacts that span the region, i.e., a low insulator score. These values showed strong concordance within each stage at 2 kb resolution (Fig. EV5A–C), and we used them to visualize compartments by defining their surrounding insulator boundaries as regions with locally minimum insulator scores across the viral genome (Fig. 6E–G). These results are consistent with previous work demonstrating that the KSHV genome is highly structured before and after reactivation (De Leo et al, 2017; Kang et al, 2011; Campbell et al, 2018, 2022). However, we note that the insulator strength in the final stage is globally reduced (Fig. EV5C), supporting reduced chromatinization of the viral genome in later stages of the lytic cycle (Toth et al, 2010). We did not observe any large shifts in 3D structures when silencing polyA75, TSSalt, or TSS72 when compared to other targets in their stages (Fig. EV5D–F), suggesting they only regulate the 3D genome structure via their regulatory targets, and that the large shifts we observe in the 3D genome structure are due to the direct or indirect activity of the viral proteins in this network.

In addition to observing changes in the compartment structure between stages, we also quantified the strength of the insulators at a given region that separates these compartments. As the regulatory circuit progresses, there is a weakening of the insulator at 100–102 kb, an appearance of an insulator at 128–130 kb, and the loss of an insulator at 136–138 kb (Fig. 6H–J). This corresponds to the region at 128–138 kb moving from a compartment at 100–130 kb to form the ~129–154 kb insulated region we originally noted (Fig. 5D). As this region contains ORF68, we next asked how these changes affect the contact frequency between the regulatory elements and their targets. While we do not see any large shifts in the physical interactions of the ORF72 locus with TSS75 relative to the reactivated control (Fig. 6G), the interaction between polyA75 and TSS75 is stronger in both the initial and intermediate stages (Fig. 6H). For example, targeting PolyA75, which in our model reduces ORF75 expression, leads to a similar effect as reducing ORF75 expression directly by targeting TSS75. This is consistent with our functional data showing that targeting polyA75 reduces ORF75 function even before reactivation (Fig. EV2D) and may represent a dilution of the interactions as the

insulator at 136–138 kb is lost (Fig. 6J) and the compartment containing polyA75 and TSS75 is expanded (Fig. 6F,G). In contrast, we find that TSSalt and TSS68 are farther apart when targeting either TSS75 or TSS57 (Fig. EV5I), suggesting that this regulatory interaction is inhibited when silencing these regions. While TSS75 likely also acts directly on TSS68 via the ORF75 protein (Fig. 3H), this allows us to complete the regulatory network by incorporating the effect of targeting TSS57 on the interaction between TSS68 and TSSalt (Fig. EV5J). Thus, by silencing individual regulatory elements and performing capture Hi-C, we determined how each element contributes to the 3D structure of the viral genome; by further incorporating the model for ORF68 transcriptional regulation, we also mapped the 3D changes to the genome as the viral lifecycle progressed.

## Discussion

Here, we identified novel regulatory events across the KSHV genome that control the expression of ORF68 and were able to distinguish between coding and noncoding elements, identify their regulatory targets, and establish their changing physical relationships in the 3D genome. Like many other viral early genes, ORF68 has a simple expression pattern: it turns on early in the lytic cycle and remains on throughout. In this light, the model regulatory network we built—and characterized through exhaustive functional genomics, transcriptional profiling, and capture Hi-C—is surprisingly complex. The virus borrows widely from the regulatory toolbox of its human host, spanning promoters, enhancers, repressors, and 3D structural elements. However, unlike the host, all these elements are contained within a compact, densely encoded genome, allowing us to identify a near-complete architecture controlling gene expression within the human nucleus.

Our functional model does not make predictions for the mechanistic nature of all these regulatory events. For example, the coding elements may or may not directly regulate transcription on the viral genome. Indeed, whereas ORF50 has been previously characterized as a viral transcription factor (Guito and Lukac, 2012; Kato-Noah et al, 2007), ORF75 more likely acts indirectly by preventing inhibition of viral transcription by host factors (Full et al, 2014). Similarly, the noncoding elements could involve ncRNAs, microRNAs, or enhancers, as CRISPRi is expected to be able to inhibit each of these; though as we see a direct effect on transcription (Fig. 2B), active enhancer marks for two regions (Fig. 4D), and physical proximity between the regulatory region and target (Fig. 5), we favor an enhancer model. Given the density of the viral genome and the width of CRISPRi footprint, it is possible a peak corresponds to multiple regulatory events or that two nearby peaks both inhibit the same regulatory locus. The TSSalt here could be an example of the former, as there are nearby K12 TSS and miRNA loci (Fig. EV1E), and indeed a recent

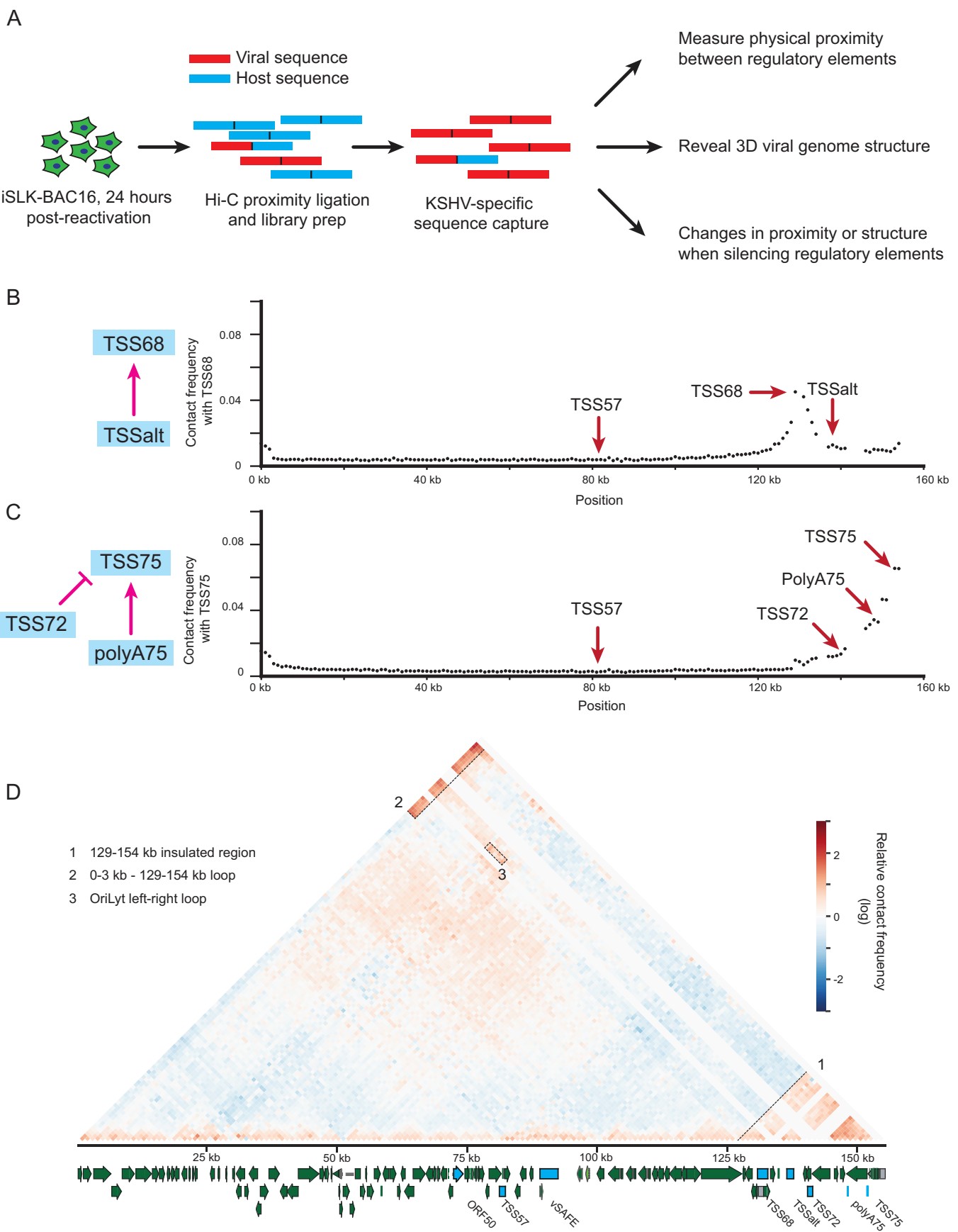

◀ **Figure 5.   Capture Hi-C of the KSHV genome.**

(A) Schematic of capture Hi-C experiment. (B, C) Contact frequency between (B) TSS68 and (C) TSS75 and other locations in the viral genome at 1 kb resolution. (D) Relative contact frequency map corrected for circular/concatenated distance. Noted features are marked and labeled. Positive values represent more interaction than expected. The annotated viral genome is provided with regulatory elements identified and marked in blue. Source data are available online for this figure.

investigation has proposed that two distinct sequences here have regulatory activity (Chowdhury et al, 2024). The polyA75 peak could be an example of the latter as it behaves very similarly to the nearby TSS75 peak. Though with the viral density and propensity for multiple functions encoded within the same locus, these issues may be intrinsic to the study of gene regulation on the viral genome, but it is possible that other functional perturbations, such as using base editors or dCas9 could allow us to pinpoint the underlying regulatory sequence.

We also examined in a low throughput manner whether this regulatory network was conserved in another, naturally infected B-cell line. Many of these components remain essential for infectious virion production (Fig. EV3B), although the results were more mixed when measuring their effects on ORF68 expression. As expected, targeting the TSS of ORFs 50, 57, or 68 reduced ORF68 expression but, in general, targeting the noncoding elements only partially recapitulated the results in iSLK cells (Fig. EV2E). This is in line with the idea that regulatory controls and, indeed, enhancer activity vary in different cell types (Andersson et al, 2014). However, there are also limitations to interpreting these results, both because the high KSHV genome copy number in BCBL1 cells and distinct TSS's and chromatin states in different cell types means that these guides may not be effective at suppressing each element.

It is also difficult to distinguish the effects of disrupting ORF57 protein and the regulatory element at TSS57. Our observation that Cas9-based disruption of ORF57 did not affect ORF68 mRNA or protein suggests that ORF57 protein does not regulate ORF68. This is in agreement with previous reports using an ORF57 deletion virus, which also indicated little regulatory effect on ORF68 (Verma et al, 2015). ORF57 protein does have a broad effect on viral transcription, but this is likely through direct effects on DNA replication components, upon which the majority (but not ORF68 (Gabaev et al, 2020)) of gene expression is partially dependent. It is possible that the TSS57 promoter contributes to ORF68 regulation independent from the ORF57 protein, as many human promoters act as enhancers for distal genes (Chen et al, 2022), but we do not observe a physical association between TSS57 and either TSS75 or TSS68 (Fig. 5B,C) or the presence of active enhancer marks (Fig. 4D). Instead, we see that targeting TSS57 causes a large disruption to the 3D conformation of the KSHV genome (Fig. 6C), leading to a reduced interaction between TSSalt and TSS68 (Fig. EV5I), potentially disrupting this enhancer-promoter relationship. However, this remains a hypothesis as we cannot disrupt the promoter and its hypothetical structural activity using CRISPRi without also disrupting the protein levels of ORF57.

Our approach demonstrates the power of combining CRISPR screening tools for the discovery of viral gene regulatory networks. Given that KSHV is a nuclear double-stranded DNA virus, it incorporates the same spectrum of regulatory mechanisms as the host—transcription factors, noncoding RNAs, enhancers, and DNA structural elements. By studying these networks on the viral genome, we can thus learn both how these regulatory mechanisms function and, ultimately, how the virus controls gene expression under diverse cellular conditions and in various cell types.

## Methods

### Reagents and tools table

| Reagent/resource | Reference or source | Identifier or catalog number |
| --- | --- | --- |
| **Experimental Models** | | |
| iSLK | Myoung and Ganem, 2011 | CVCL_B6YU |
| Lenti-X 293 T | Takara | 632180 |
| TREx-RTA-BCBL1 | Nakamura et al, 2003 | CVCL_0165 |
| CRISPRi+ BCBL1s | Brackett et al, 2021 | CVCL_0165 |
| KSHV BAC16 | Brulois et al, 2012 | N/A |
| **Recombinant DNA** | | |
| pMD2.G | Addgene | # 12259 |
| pMDLg/pRRE | Addgene | # 12251 |
| pRSV-Rev | Addgene | # 12253 |
| pMCB320 | Addgene | # 89359 |
| lentiCas9-Blast | Addgene | # 52962 |
| Lenti-dCas9-KRAB-blast | Addgene | # 89567 |
| **Antibodies** | | |
| Rabbit anti-CTCF | Cell Signaling | D31H2 |
| Rabbit anti-H3K27ac | Cell Signaling | D5E4 |
| Rabbit anti-H3K4me1 | Cell Signaling | D1A9 |
| **Oligonucleotides and other sequence-based reagents** | | |
| Primer and guide sequences | This study | Dataset EV6 |
| KSHV guide library | Morgens et al, 2022 | Addgene #180272 |
| **Chemicals, enzymes, and other reagents** | | |
| DMEM | Gibco | 11965-118 |
| Pen-strep | Gibco | 15140-122 |
| Glutamax | Gibco | 35050079 |
| FBS | Peak Serum | N/A |
| Hygromycin B | Gibco | 10687010 |
| Blasticidin | Thermo | A1113903 |
| G418 | VWR | 97063-060 |
| Puromycin | Thermo | A1113803 |
| Trypsin | Gibco | 25300-120 |
| Polyethylenimine | Polysciences | 23966 |
| JF646 HaloLigand | Promega | GA1121 |
| PFA | Pierce | PI28906 |
| Protease K | Promega | EO0491 |
| QIAamp DNA Blood Mini Kit | Qiagen | 51104 |
| Doxycycline hyclate | Millipore | D9891 |

| Reagent/resource | Reference or source | Identifier or catalog number |
|---|---|---|
| Sodium Butyrate | Sigma | 303410 |
| KAPA RNA Hyper Prep kit | Roche | KK8581 |
| KAPA Library Quantification Kit | Roche | KK4824 |
| RNeasy Plus Micro kit | Qiagen | 74034 |
| DNAase I | Lucigen | QER090150 |
| AMV RT | Promega | M5108 |
| RNasin | Promega | N2515 |
| iTaq Universal SYBR Green | BioRad | 1725122 |
| 5-ethynyl uridine | Invitrogen | C10365 |
| Click-IT Nascent RNA Capture Kit | Invitrogen | C10365 |
| SuperScript VILO cDNA Synthesis Kit | Thermo | 11754050 |
| PowerUP SYBR Green Master Mix | Thermo | A25778 |
| MinElute PCR Purification Kit | Qiagen | 28004 |
| GeneJET™ Gel Extraction Kit | Thermo | ab11826 |
| Polyjet In Vitro Transfection Reagent | SignaGen | NC1536117 |
| Polybrene | Fisher | TR1003G |
| Spermidine | Sigma | S2626 |
| EDTA-free protease inhibitor cocktail | Sigma | 5056489001 |
| Concanavalin A | Sigma | C0412 |
| pAG-MNase | Cell Signaling | 40366S |
| RNAase A | Thermo | EN0531 |
| Spike-in DNA | Cell Signaling | 40366S |
| Protease K | Cell Signaling | 10012 |
| NEXT UltraII DNA Library Prep | NEB | E7645L |
| Unique Dual Index UMI Adaptors | NEB | E7395S |
| AMPure XP | Beckman | A63880 |
| Hi-C+ kit w/custom enrichment panel | Arima Genomics | A311027 |
| Arima Library Prep Module | Arima Genomics | A303011 |
| **Software** | | |
| MultiQC | N/A | N/A |
| HTStream 1.3.0 | N/A | N/A |
| STAR 2.7.1a | N/A | N/A |
| Spyder 5.3.3 | N/A | N/A |
| Cutadapt | Martin, 2011 | N/A |
| Bowtie | Langmead et al, 2009 | N/A |
| Bowtie 2 | Langmead and Salzberg, 2012 | N/A |
| Umi_tools | Smith et al, 2017 | N/A |
| bamCoverage | Ramírez et al, 2016 | N/A |

| Reagent/resource | Reference or source | Identifier or catalog number |
|---|---|---|
| HiCUP 0.8.0 | Wingett et al, 2015 | N/A |
| CHICAGO | Cairns et al, 2016 | N/A |
| **Other** | | |
| Aria II | BD | N/A |
| NovaSeq 6000 | Illumina | N/A |
| Accuri C6 Plus | BD | N/A |
| 2100 Bioanalyzer | Agilent | N/A |
| Fragment Analyzer | Agilent | N/A |
| CFX Connect | BioRad | N/A |
| Quantstudio 3 | Thermo | N/A |
| NextSeq 2000 | Illumina | N/A |

## Plasmid and oligos

pMD2.G (Addgene plasmid # 12259), pMDLg/pRRE (Addgene plasmid # 12251), and pRSV-Rev (Addgene plasmid # 12253) were gifts from Didier Trono. pMCB320 was a gift from Michael Bassik (Addgene plasmid # 89359). lentiCas9-Blast was a gift from Feng Zhang (Addgene plasmid # 52962). Lenti-dCas9-KRAB-blast was a gift from Gary Hon (Addgene plasmid # 89567). Sequences used are listed in Dataset EV6.

## Cell culture and plasmids

iSLK (Myoung and Ganem, 2011) and Lenti-X 293T (Takara) cells were maintained in DMEM (Gibco +glutamine, +glucose, -pyruvate) with pen-strep (Gibco; 100 I.U./mL) and 1X Glutamax (Gibco) along with 10% FBS (Peak Serum). TREx-RTA-BCBL1(Nakamura et al, 2003) were maintained in RPMI (Gibco +glutamine) with pen-strep (Gibco; 100 I.U./mL) and 1X Glutamax (Gibco) along with 20% FBS (Peak Serum). Cell lines were neither authenticated nor recently tested for mycoplasma. iSLK cells were maintained in 1 ug/mL puromycin, 50 ug/mL G418, and 125 ug/mL hygromycin B (Gibco). Cas9+ and CRISPRi + cells were additionally maintained in 10 ug/mL blasticidin. 0.05% Trypsin (Gibco) was used to passage cells. All cells were maintained at 37 °C and 5% $CO_2$ in a humidity-controlled incubator. Lenti-X 293T cells were obtained from the UCB Cell Culture Facility.

## Generation of CRISPRi Halo-ORF68 iSLK line

iSLK cells latently infected with a copy of BAC16 (Brulois et al, 2012) containing a HaloTag-ORF68 fusion at the endogenous locus were lentivirally infected with dCas9-KRAB (CRISPRi). Briefly, Lenti-X cells were transfected with third-generation lentiviral mix (pMDLg/pRRE, pRSV-REV, and pMD2.G) and dCas9-KRAB (blastR) with polyethylenimine (Polysciences). The supernatant was harvested at 48 and 72 h and 0.45 um filtered before applying to iSLK cells for 48 h. This process was then repeated to ensure high CRISPRi expression.

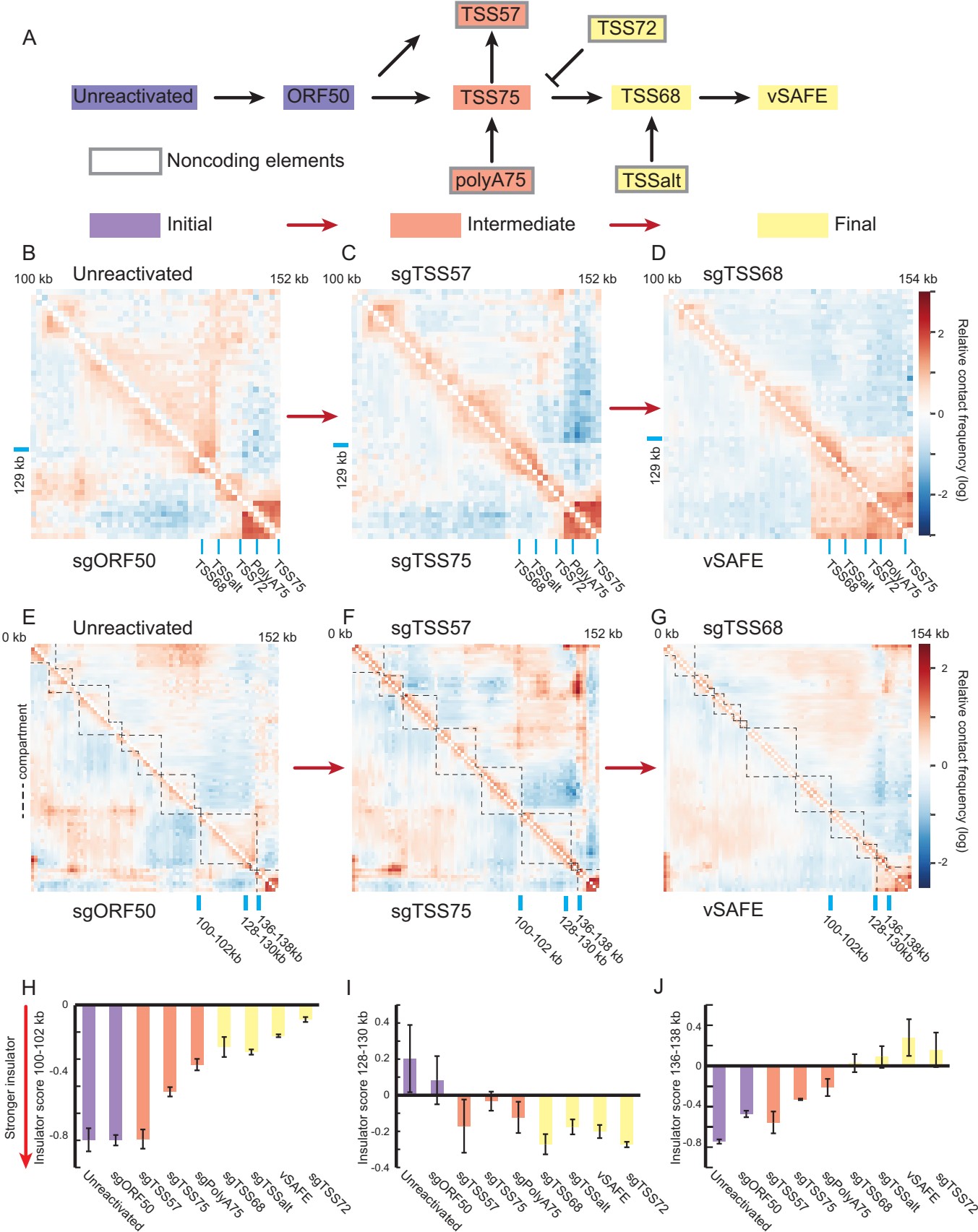

**Figure 6.   Changing physical relationship between regulatory regions.**

(A) Schematic splitting of the regulatory network into initial, intermediate, and final stages. (B–D) Relative contact frequency at 1 kb resolution for the viral region from 100–154 kb measured by capture Hi-C 24 h post-reactivation for representative (B) initial stages (purple), (C) intermediate stages (orange), (D) and final stages (yellow). Positive values indicate greater interaction than expected, and locations of regulatory elements are marked in blue. (E–G) Relative contact frequency at 2 kb resolution across the genome measured by capture Hi-C 24 h post-reactivation for representative (E) initial stages, (F) intermediate stages, (G) and final stages. Dotted lines represent locations of insulator regions as defined by negative local-minimum insulator scores. The locations of the three observed insulators are marked in blue. (H–J) Insulator scores at the marked locations in (A–C) at (D) 100–102 kb, (E) 130–132 kb, and (F) 136–138 kb. More negative values indicated a stronger insulator. Values are the average of two adjacent regions with error bars representing the standard deviation of these two values. Source data are available online for this figure.

## CRISPRi screening and analysis

A library of guides tiling the KSHV BAC16 genome was delivered lentivirally to the CRISPRi+ iSLK cells above. Four days later, cells were reactivated with 5 ug/mL doxycycline and 1 mM sodium butyrate and treated with 10 nM JF646 Haloligand (Promega). Twenty-four hours post reactivation, cells were fixed in 4% PFA, and sorted for high and low ORF68 expression using a BD Aria II. Cells were then unfixed overnight in 150 mM NaCl, 60 C, and 50 ug/mL protease K (Promega). DNA was then extracted using a single column of QIAamp DNA Blood Mini Kit (Qiagen), following the manufacturer's protocol and adjusting the initial reaction volume. The sgRNA locus was then amplified, and library adapters ligated as previously described (Deans et al, 2016). Libraries were sequenced on an Illumina NovaSeq 6000.

Counts for individual guides were converted to enrichment scores by calculating the log2 ratio of counts between high and low populations relative to the median negative control. Enrichment values from two replicates were averaged. To calculate the significance of a given window, a 500 bp sliding window was used, comparing the enrichment of each guide and a 500 bp neighborhood to the enrichment scores of all negative controls using a Mann–Whitney test. An arbitrary $p$ value cutoff was used to identify peaks.

## Individual guide delivery

For each pool of sgRNAs, independent lentiviruses were produced as above, then pooled and applied to CRISPRi-positive HaloTag-ORF68 iSLK cells for 48 h. For protein analysis, cells were reactivated in the presence of 10 nM JF646 Haloligand (Promega) with doxycycline and sodium butyrate as above. Cells were then analyzed for fluorescence at 24 and 48 h post-reactivation from four independent reactivations on a BD Accuri C6 plus.

For latency analysis, cell lines were maintained in triplicate with blasticidin, puromycin, and G418 but in the absence of hygromycin. Cells were split every 48 h and GFP levels were measured by BD Accuri C6 Plus. The percent of GFP-positive cells on day 11 was normalized to the percent positive on day 1, and this ratio was again normalized to the loss of GFP observed in parental cells without a guide RNA.

## RNA-seq and analysis

RNA samples from above were sent for library preparation and sequencing at the QB3-Berkeley Genomics core labs (RRID:SCR_022170). Total RNA quality, as well as poly-dT enriched mRNA quality, were assessed on an Agilent 2100 Bioanalyzer. Libraries were prepared using the KAPA RNA Hyper Prep kit (Roche KK8581). Truncated universal stub adapters were ligated to cDNA fragments, which were then extended via PCR using unique dual indexing primers into full-length Illumina adapters. Library quality was checked on an AATI (now Agilent) Fragment Analyzer. Library molarity was measured via quantitative PCR with the KAPA Library Quantification Kit (Roche KK4824) on a BioRad CFX Connect thermal cycler. Libraries were then pooled by molarity and sequenced on an Illumina NovaSeq 6000 S4 flowcell for 2 × 150 cycles, targeting at least 25 M reads per sample. Fastq files were generated and demultiplexed using Illumina bcl_convert and default settings, on a server running CentOS Linux 7.

Sequencing quality was assessed with MultiQC, and reads were preprocessed with HTStream version 1.3.0 including deduplication. Genome indices were prepared using STAR 2.7.1a. The human GRCh38.p13 genome assembly was indexed with Gencode v43 annotations. Due to overlapping transcripts on the KSHV BAC16 genome, individual exon coordinates were assigned to the corresponding parent transcript. Preprocessed reads were then aligned using STAR, and count files were generated for transcripts. Any transcript with no reads in all replicates and conditions was eliminated from further analysis. Reads from *E.coli* genes on the BAC were also removed. Raw viral counts were normalized to total input reads for each sample and subsequently normalized to within replicate vSafe condition values. Correlations and heatmaps were generated using the matplotlib, pandas, and seaborn packages in Spyder 5.3.3.

## RT-qPCR and EU-RT-qPCR

For RT-qPCR analysis, RNA was extracted at 24 h post-reactivation using an RNeasy Plus Micro kit (Qiagen), treated with DNAase I (Lucigen), and reverse transcribed using AMV RT (Promega) and random 9-mers in the presence of RNasin (Promega). qPCR was then performed on a Quantstudio 3 using the indicated targets with iTaq Universal SYBR Green (BioRad). RQ values were calculated using a standard curve. Results are from four independent reactivations.

For nascent RNA measurements, cells were reactivated, and 24 h post-reactivation were treated with 200 uM 5-ethynyl uridine (EU) for two hours. RNA was then extracted using the RNeasy Micro Plus Kit (Qiagen), and EU-incorporated RNA was labeled and purified per the Click-iT™ Nascent RNA Capture Kit (Invitrogen). Reverse transcription was performed on-bead using SuperScript™ VILO™ cDNA Synthesis Kit (Thermo) reverse transcriptase, and qPCR was performed with iTaq Universal SYBR Green (BioRad) or PowerUp™ SYBR™ Green Master Mix (Thermo) on a Quantstudio 3

(Thermo). Relative quantities were calculated using a standard curve. Results are from three independent reactivations.

## CRISPR nuclease screen and analysis

Cas9 screen was performed as the CRISPRi screen above using a Cas9+ iSLK cell line infected with a copy of BAC16 containing a HaloTag-ORF68 fusion. The library was then amplified using staggered primers (Tsui et al, 2023) and a modified amplification protocol previously described (Morgens et al, 2019). qPCR was used to determine the cycle number at ¼ CT. All PCR product was run over a single Minelute column (Qiagen) for each sample. Size selection was performed using a gel extraction (Thermo). The library was sequenced on Illumina NextSeq 2000 with a 150 bp single-read using Illumina sequencing primers. Adapters were removed using cutadapt (Martin, 2011), and reads were aligned to the library using bowtie (Langmead et al, 2009).

Log2 values were calculated as above, and enrichment values were averaged from two replicates. For each exon boundary, a window of 500 bp on one side was tested against a 500 bp on the other to calculate $p$ values using the Mann–Whitney test. Median values were used to determine the sign of the shift. Exons with $p$ values <0.001 and consistent signs were considered hits.

## Transfection of ORF68 reporter

HEK293T cells were transfected using PolyJet In Vitro DNA Transfection Reagent (SignaGen) with a plasmid containing 240 basepairs upstream of ORF68's start codon (Gardner and Glaunsinger, 2018) driving a HaloTag. Cells were cotransfected with either a plasmid expressing ORF75 or ORF50. 24 h later, cells were treated with JF646 Halo Ligand (Promega) for 24 h before analysis on a BD Accuri C6 Plus. Cells were then gated for the JF646 signal, and the average fluorescence for the JF646 positive cells was calculated. Data are from seven independent replicates.

## CRISPRi, RT-qPCR, and supernatant transfer in BCBL1s

CRISPRi+ BCBL1s (Brackett et al, 2021) were generously provided by Dr. Carolina Arias and Dr. Mark Manzano, and maintained in RPMI with additional glutamax, penn/strep, and 20% FBS along with 10 ug/mL Blasticidin. Cells were spinfected in the presence of 8 ug/mL polybrene with pools of lentiviral guides and selected using 1 ug/mL puromycin for 1 week. Cells were then reactivated with 5 ug/mL doxycycline, and 24 h later, RNA was extracted, and RT-qPCR was performed as above. For supernatant transfer, 96 h post reactivation, the supernatant was filtered using a 0.45-um filter and incubated with naïve HEK293T cells for 24 h, followed by DNA extraction using Qiagen Blood Mini and qPCR for viral DNA as above.

## CUT&RUN

CUT&RUN was performed using a modified protocol (Skene et al, 2018; Miura and Chen, 2020) using antibodies against CTCF (D31H2 rabbit; Cell Signaling), H3K27ac (D5E4 rabbit; Cell Signaling), and H3K4me1 (D1A9 rabbit; Cell Signaling). Cas9+ iSLK-BAC16 cells were infected with guide RNAs targeting ORF75, ORF50, or negative controls as above. Cells were reactivated using sodium butyrate and doxycycline as above. 24 h post reactivation, cells were trypsinized and nuclei were isolated in permeabilization buffer (20 mM HEPES-KOH pH 7.5; 150 mM NaCl; 0.1% Triton X-100; 0.5 mM Spermidine; EDTA-free protease inhibitor cocktail (Sigma)). Nuclei were then spun onto plates pretreated with Concanavalin A (Sigma) and washed in a permeabilization buffer. Nuclei were then incubated at RT in primary antibody in permeabilization buffer for 45 min before being washed twice in permeabilization buffer. Nuclei were then incubated at RT in pAG-MNase (Cell Signaling) in permeabilization buffer for 45 min before being washed twice in permeabilization buffer. Nuclei were then incubated in 5 mM calcium chloride in permeabilization buffer at 4 °C for 30 min. 4X Stop Buffer (680 mM NaCl; 40 mM EDTA; 8 mM EGTA; 0.1% Triton X-100; 100 ug/mL RNase A (Thermo)) containing spike-in DNA (Cell Signaling) was then added, and nuclei were incubated at 37 °C for 30 min. The supernatant was then collected and incubated for >1 h at 50 °C after the addition of 0.2% SDS and 12.5 ug/mL Proteinase K (Cell Signaling). DNA then was extracted using phenol-chloroform.

Libraries were then prepped using NEB NEXT UltraII DNA library prep (NEB) with unique dual index UMI adapters (NEB). Pre-PCR size selection was altered from protocol to include two 1.1X AMPure XP (Beckman) clean-ups. qPCR was used to determine cycle numbers. Post-PCR size selection using AMPure XP beads was altered to include 0.4x right-sided clean-up for histones and 0.6x right-sided clean-up for CTCF, followed by two 1.1x left-side clean-up for all samples. Samples were then sequenced on a 50 PE NextSeq 2000. Sequencing reads were then trimmed with CUTADAPT (Martin, 2011) and aligned separately to the BAC16 genome and the spike-in yeast genome using bowtie2 (Langmead and Salzberg, 2012) with the following options: –end-to-end –no-mixed –no-discordant -I 10 -X 700 –dovetail. Reads were then deduplicated using UMIs and umi_tools (Smith et al, 2017). Bedgraphs were then made using bamCoverage (Ramírez et al, 2016), normalizing to the total reads aligned to the spike-in, and then averaged from three replicates. Nucleosomal fragments (>120 bp) were removed from CTCF samples, and sub-nucleosomal fragments (<120 bp) were removed from histone samples.

## Supernatant transfer

Cells were reactivated with 5 ug/mL doxycycline and 1 mM sodium butyrate. Seventy-two hours post reactivation, supernatant was filtered through a 0.45-um PES filter and applied to naïve HEK293T cells for 24 h. HEK cells were then counted on an Accuri C6 Plus (BD) and percent GFP positive was used to calculate infection.

## BAC16-specific capture Hi-C

Capture Hi-C data was generated using the Arima-Hi-C+ kit, a custom Arima panel specific to the BAC16 KSHV genome, and the Arima Library Prep Module according to the Arima Genomics manufacturer's protocols. Briefly, 24 h post reactivation, cells were fixed in 4% PFA and frozen. Chromatin was then purified, digested, and filled-in with biotin-labeled nucleotides. Proximity ligation was performed, DNA was purified and sheared, and ligated molecules were enriched by biotin pulldown. Library preparation was then performed, and viral-specific sequences were enriched using

capture probes. Libraries were then sequenced using a NextSeq 2000 P2 150PE kit (Illumina).

## Capture Hi-C analysis

Ditags for Hi-C analysis were aligned to the BAC16 KSHV genome, including the HaloTag insertion, and filtered using HiCUP version 0.8.0 (Wingett et al, 2015) and converted to fragment level counts using CHICAGO bam2chicago (Cairns et al, 2016). Fragments corresponding to the terminal repeats were removed before analysis. The viral genome was then split into bins of 0.5–5 kb, and each fragment was assigned to a bin based on the center of the fragment. Each bin was removed if it had less than a total of 5000 associated ditags. Bin level counts were then normalized to a symmetric stochastic matrix to calculate contact frequencies by iterative normalization of both rows and columns (Sinkhorn-Knopp). Distance to contact frequency relationship was then graphed using a linear distance metric, and a power law was fit. A second distance metric assuming a circular or concatenated genome was then used, allowing for a 180 kb genome size including ~30 terminal repeats; if the linear distance was greater than half the genome size, then the genome size minus the linear distance was used. A power law was fit to this distribution, and relative contact frequencies are represented as the log ratio of the observed vs the expected power law distribution.

Insulator scores were calculated using the principles presented in Crane et al, 2015 (Crane et al, 2015). Contact frequencies spanning each region at 6–10 kb were averaged and normalized to the median value across the genome. Local minima of values lower than 0 were used as insulator locations.

# Data availability

The datasets produced in this study are available in the following databases: Raw sequencing reads for CRISPRi screens, Cas9 screens, RNA-seq, CUT&RUN, and capture Hi-C: Bioproject PRJNA1151019 (https://www.ncbi.nlm.nih.gov/bioproject/?term=PRJNA1151019). Processed count files for screens, processed viral counts for RNA-seq, bedgraphs for processed CUT&RUN, and processed counts for capture Hi-C: Dataset EV1-8.

The source data of this paper are collected in the following database record: biostudies:S-SCDT-10_1038-S44320-024-00075-0.

# Peer review information

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

## Acknowledgements

The authors would like to thank the members of the Glaunsinger lab for their feedback and support, and Dr. C. Kimberly Tsui for the stagger primer sequences. Cell lines were obtained from the UCB Cell Culture Facility, which is supported by The University of California Berkeley (SCR_017924). RNA-seq was performed by the Berkeley functional genomics core, and sequencing was performed through the Vincent J. Coates Genomics Sequencing Laboratory at UC Berkeley (QB3 Genomics, UC Berkeley, Berkeley, CA, RRID:SCR_022170). Flow cytometry and FACS were conducted at the CRL Flow Cytometry Facility. We thank Hector Nolla and Alma Valeros of the UC Berkeley Cancer Research Laboratory Flow Cytometry Facility for their training and expertise. BAG is an investigator of the Howard Hughes Medical Institute, and DWM was a Howard Hughes Medical Institute Awardee of the Life Sciences Research Foundation. This research was also funded by NIH grant AI122528 and R01 CA136367 to BAG and 1K99AI173531-01A1 to DWM.

## Author contributions

**David W Morgens**: Conceptualization; Software; Formal analysis; Funding acquisition; Investigation; Visualization; Methodology; Writing—original draft; Writing—review and editing. **Leah Gulyas**: Software; Formal analysis; Investigation; Visualization; Methodology; Writing—review and editing. **Xiaowen Mao**: Formal analysis; Investigation; Methodology. **Alejandro Rivera-Madera**: Investigation; Methodology. **Annabelle S Souza**: Investigation. **Britt A Glaunsinger**: Conceptualization; Supervision; Funding acquisition; Writing—original draft; Writing—review and editing.

Source data underlying figure panels in this paper may have individual authorship assigned. Where available, figure panel/source data authorship is listed in the following database record: biostudies:S-SCDT-10_1038-S44320-024-00075-0.

## Disclosure and competing interests statement

The authors declare no competing interests.

# Expanded View Figures

**Figure EV1. Supplementary screen data.**

(**A**) Enrichment of individual guides at the ORF68 locus. Each dot represents a single guide, with the target location displayed on the x-axis and the average enrichment from two replicates on the y-axis. (**B**) Reproducibility of guide enrichments from two replicates. (**C–H**) Smoothed enrichment of guides at an indicated locus with annotated transcription start sites (Ye et al, 2019). For each guide, the median enrichment of a 100 bp window centered at the target locus was calculated along with an interquartile range (IQR) to represent the range of values. The median value is shown as a point, and IQR is shown as a shaded region. Regions significant ($p < 10^{-11}$) in sliding window analysis are shown in blue. NA indicates unannotated TSSs.

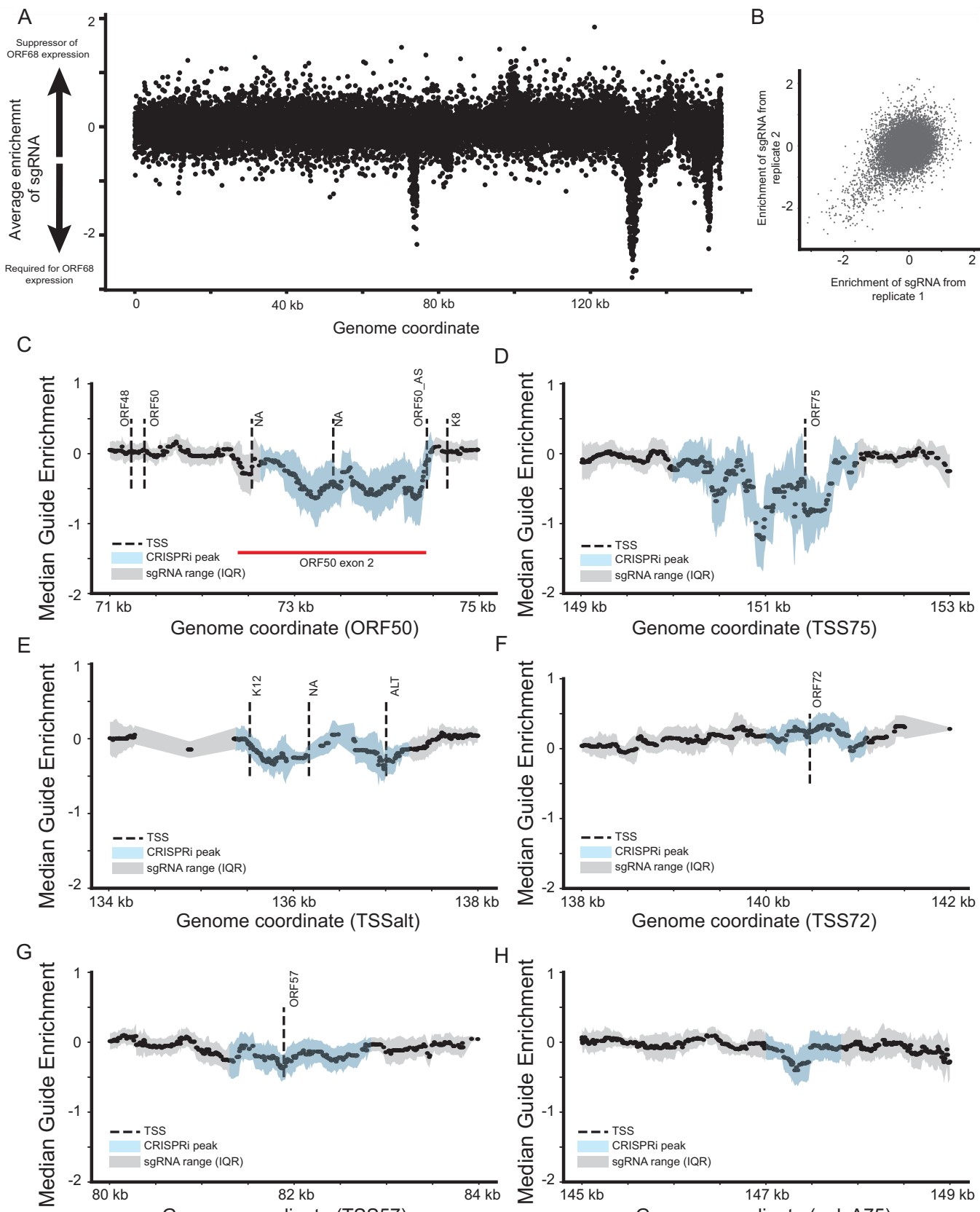

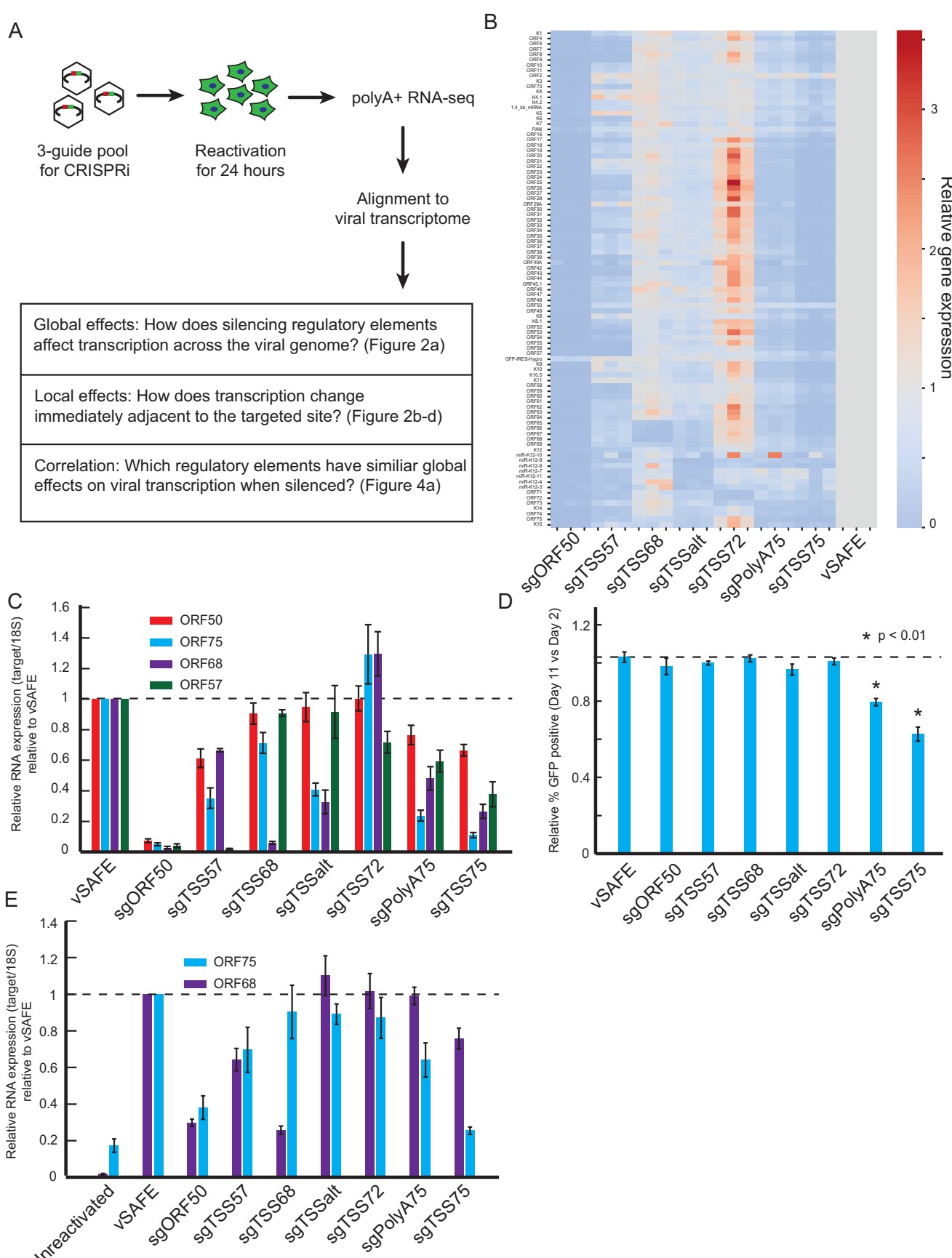

**Figure EV2. Supplementary for RNA-seq.**

(A) Schematic showing set-up of RNA-seq experiment on CRISPRi cells infected with a three-guide pool, reactivated, and polyA+ RNA-seq at 24 h post-reactivation. (B) Heatmap indicating viral gene expression relative to the matched vSAFE replicate of each individual replicate. Rows are presented in genome order. Replicates from three independent reactivations. (C) RT-qPCR was used to measure how CRISPRi-based repression of the individual elements indicated on the x-axis influenced the levels of ORFs 50, 75, 68, and 57 mRNA. Error bars represent standard error centered on the mean of four independent reactivations. (D) Effect on latency measured by loss of virally encoded EGFP expression over 10 days. Mean values are presented relative to parental cells, and error bars are standard errors from three parallel replicates. $P$ values are calculated by $t$-test; exact $p$ values for marked values are 0.0031 for sgPolyA75 and 0.0013 for sgTSS75. (E) RT-qPCR of viral genes ORF75 and ORF68 in CRISPRi+ BCBL1 cells targeted with indicated guide RNAs at 24 h post-reactivation. Mean values are presented relative to 18S and vSAFE cells. Error bars are standard errors from five independent reactivations.

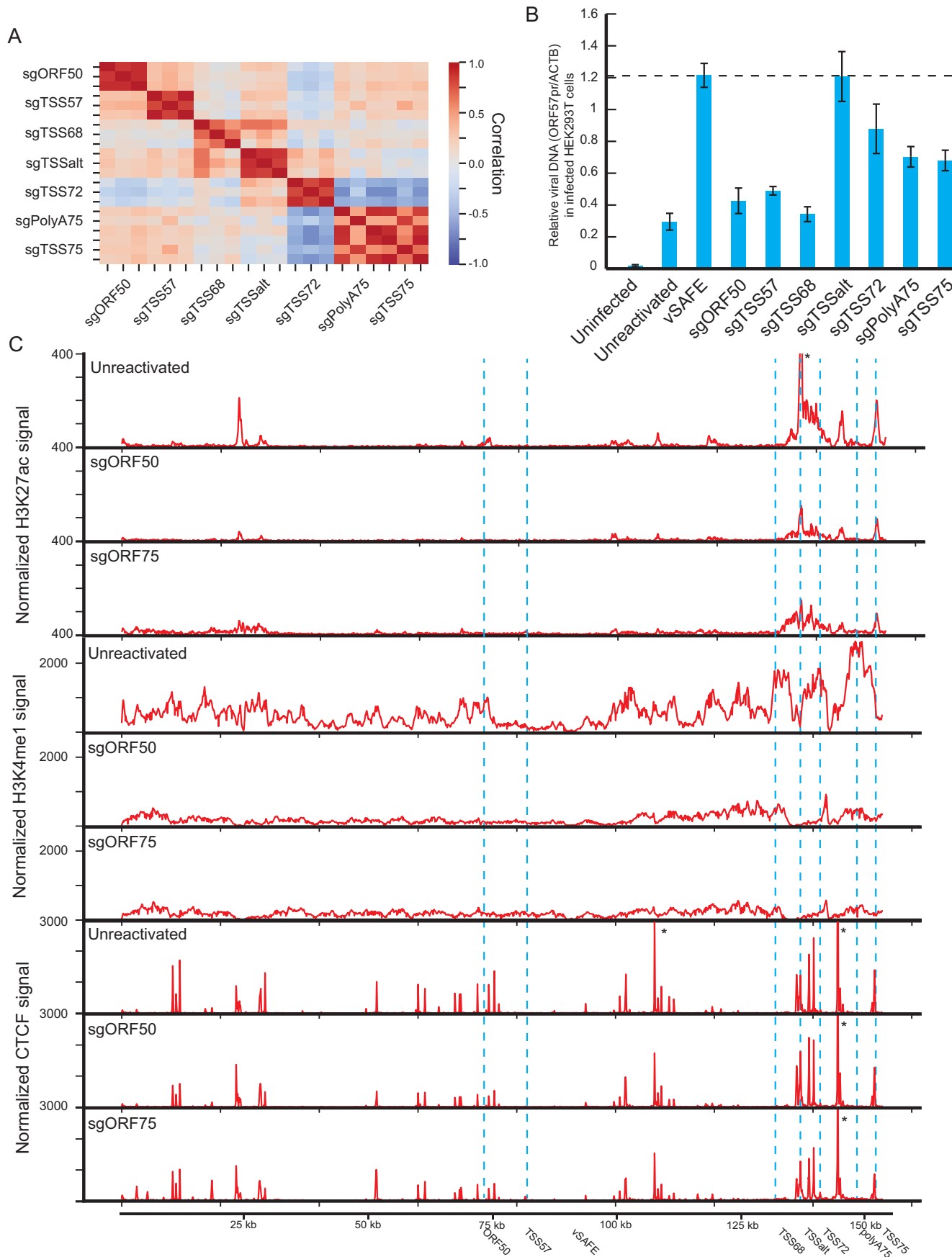

◀ **Figure EV3. Supplementary mapping data.**

(**A**) Individual replicate correlation among RNA-seq. (**B**) Supernatant transfer assay measuring changes in KSHV virion production after knockdown of the indicated loci in CRISPRi+ BCBL1 cells. qPCR measurements of viral DNA content relative to host DNA content. Error bars represent standard error centered on the mean from four independent reactivations. (**C**) CUT&RUN signal from indicated mark in reactivated iSLK cells 24 h post-reactivation. Asterisks indicate peaks above the maximum signal graphed. The signal is averaged from three independent replicates.

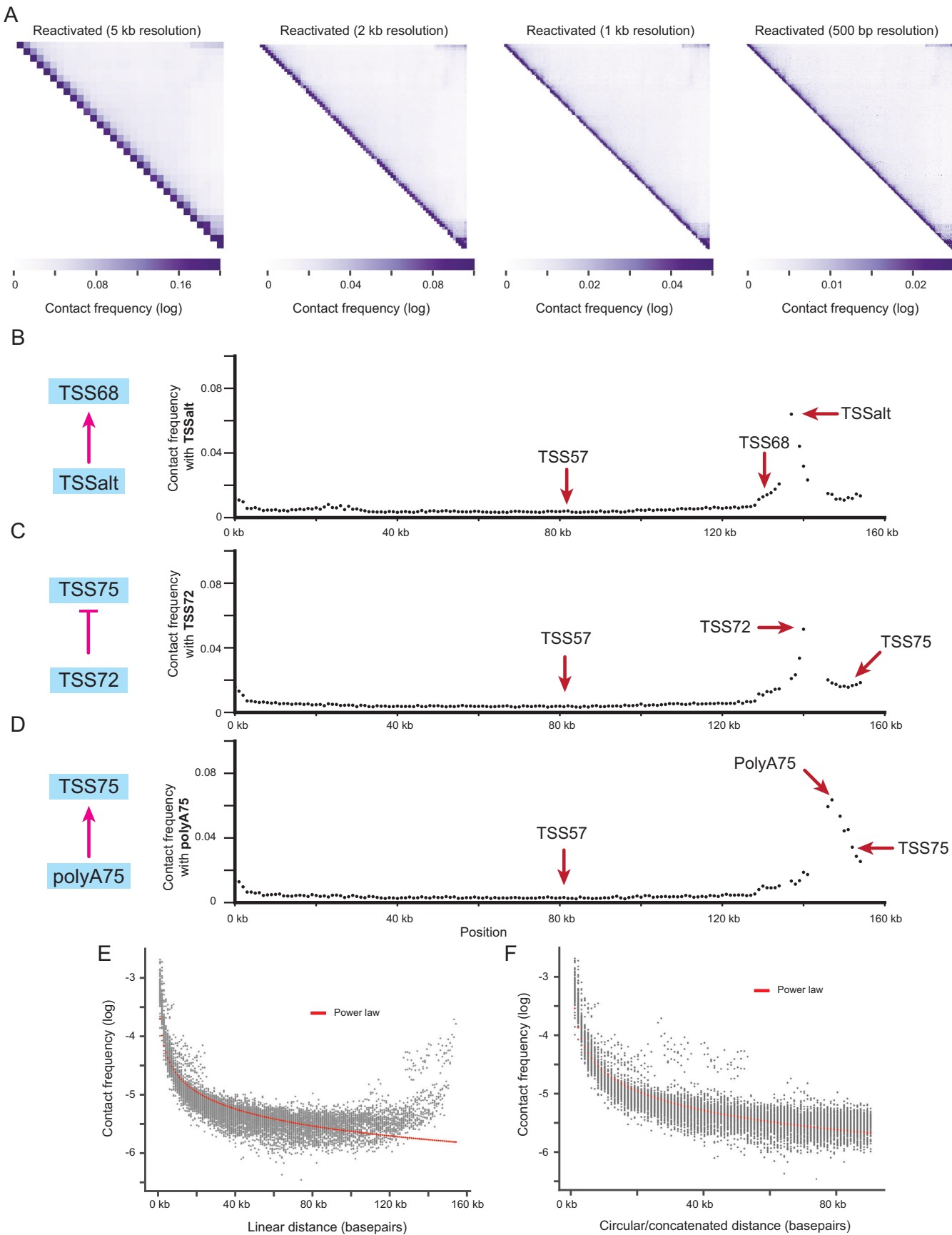

◀ **Figure EV4. Hi-C data supplement.**

(A) Contact frequency of reactivated sample at 5 kb, 2 kb, 1 kb, and 500 bp resolution. (B–D) Contact frequency between (B) TSSalt, (C) TSS72, and (D) polyA75 and other locations in the viral genome at 1 kb resolution. (E, F) The observed relationship between observed contact frequency and distance between regions when calculated using (E) a linear distance metric or (F) a circular/concatenated distance metric.

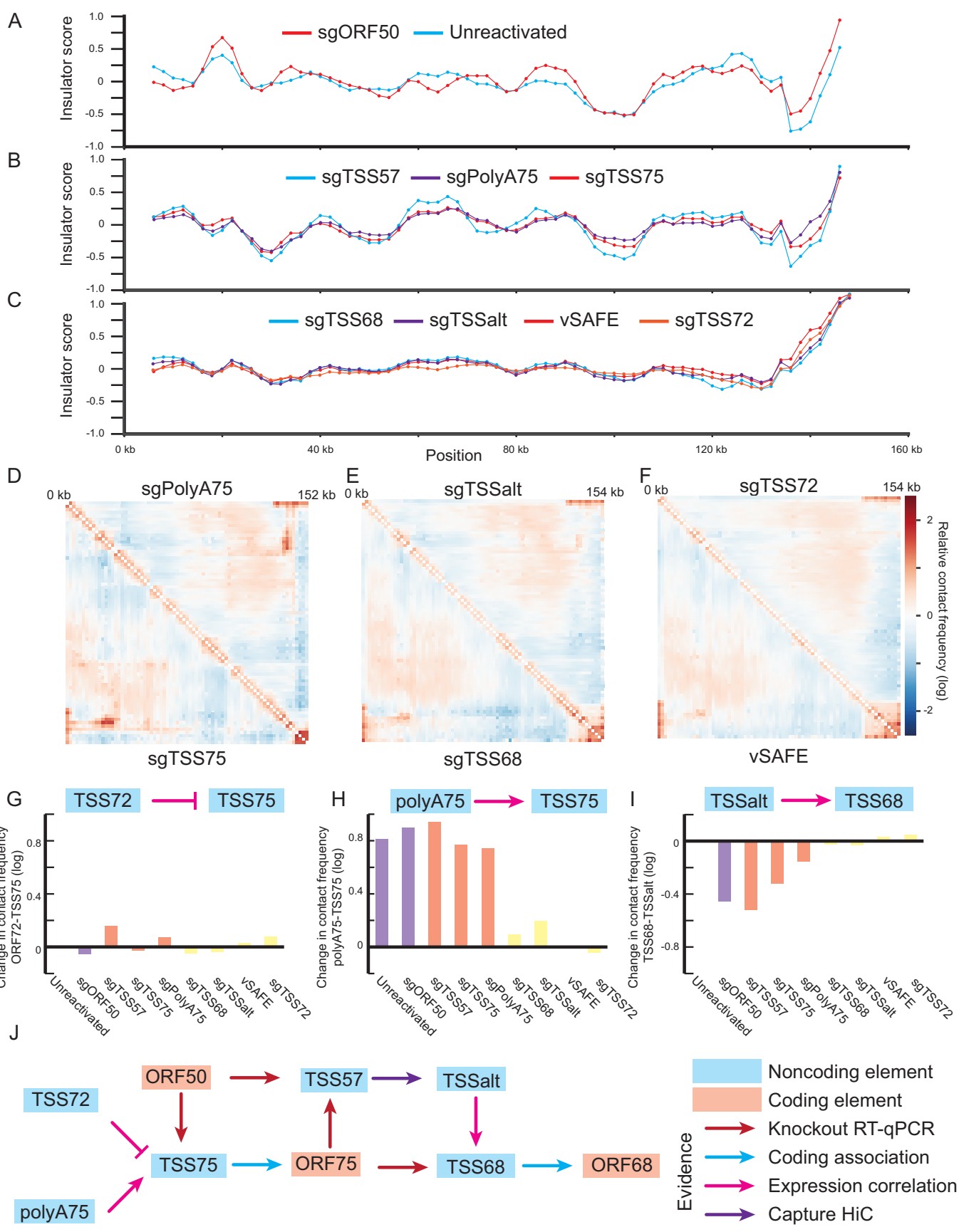

◄ **Figure EV5. Structural changes supplement.**

(**A–C**) Insulator scores for marked conditions measuring the relative frequency of reads crossing a given location. A more negative value indicates a strong insulator. Local minima were used to define the regions marked in Fig. 6E–G. (**D–F**) Relative contact frequency at 2 kb resolution for (**D**) sgPolyA75 and sgTSS75, (**E**) sgTSSalt and sgTSS68, and (**F**) sgTSS72 and vSAFE samples. Positive values indicate greater interaction than expected. (**G–I**) Change in contact frequency from reactivated cells between (**G**) TSS72 and TSS75, (**H**) polyA75 and TSS75, and (**I**) TSS68 and TSSalt at 2 kb resolution. (**J**) The final model for a regulatory circuit is based on all available data.

