## [Peer Review File · Molecular Systems Biology]

Enhancers and genome conformation provide complex transcriptional control of a herpesviral gene

David Morgens, Leah Gulyas, Xiaowen Mao, Alejandro Rivera-Madera, Annabelle Souza, and Britt Glaunsinger

Corresponding author(s): Britt Glaunsinger (glaunsinger@berkeley.edu) , David Morgens (dmorgens@berkeley.edu)

Review Timeline:

Submission Date:	8th May 24
Editorial Decision:	7th Jun 24
Revision Received:	23rd Sep 24
Editorial Decision:	24th Oct 24
Revision Received:	28th Oct 24
Accepted:	31st Oct 24

Editor: Poonam Bheda

Transaction Report:

7th Jun 2024

Manuscript Number: MSB-2024-12411

Title: From enhancers to genome conformation: complex transcriptional control underlies expression of a single herpesviral gene

Dear Dr. Glaunsinger,

Thank you again for submitting your work to Molecular Systems Biology. We have now heard back from the three reviewers who agreed to evaluate your study. As you will see from the reports below, the referees acknowledge the interest of the study and are overall supportive of your work; however they also comment on multiple aspects of the manuscript that should be strengthened in a revision.

Without repeating all the comments listed below, some of the more fundamental issues raised are the following:

- Putative regulatory loci should be further validated by investigating chromatin state via e.g. epigenetic marks, as raised by all 3 reviewers
- Key results should be validated in naturally infected cells, as iSLK cells are artificial, as raised by Reviewer 1

All other issues raised would need to be satisfactorily addressed. Please let me know in case you would like to discuss in further detail any of the comments, I would be happy to schedule a call.

We require:

1) A .docx formatted version of the manuscript text (including legends for main figures, EV figures and tables). Please make sure that the changes are highlighted to be clearly visible. Alternatively you may choose to submit your manuscript as a LaTeX file.

4) A .docx formatted letter INCLUDING the reviewers' reports and your detailed point-by-point responses to their comments. As part of the EMBO Press transparent editorial process, the point-by-point response is part of the Peer Review File (PRF), which will be published alongside your paper.

5) A complete author checklist, which you can download from our author guidelines (<https://www.embopress.org/page/journal/17574684/authorguide#submissionofrevisions>). Please insert information in the checklist that is also reflected in the manuscript. The completed author checklist will also be part of the PRF.

6) Please note that all corresponding authors are required to supply an ORCID ID for their name upon submission of a revised manuscript.

7) It is mandatory to include a 'Data Availability' section after the Materials and Methods. Before submitting your revision, primary datasets produced in this study need to be deposited in an appropriate public database, and the accession numbers and database listed under 'Data Availability'. Please remember to provide a reviewer password if the datasets are not yet public (see <https://www.embopress.org/page/journal/17574684/authorguide#dataavailability>).

This study includes no data deposited in external repositories.

8) For data quantification: please specify the name of the statistical test used to generate error bars and P values, the number (n) of independent experiments (specify technical or biological replicates) underlying each data point and the test used to calculate p-values in each figure legend. The figure legends should contain a basic description of n, P and the test applied. Graphs must include a description of the bars and the error bars (s.d., s.e.m.). Please provide exact p values.

9) Our journal encourages inclusion of *data citations in the reference list* to directly cite datasets that were re-used and obtained from public databases. Data citations in the article text are distinct from normal bibliographical citations and should

directly link to the database records from which the data can be accessed. In the main text, data citations are formatted as follows: "Data ref: Smith et al, 2001" or "Data ref: NCBI Sequence Read Archive PRJNA342805, 2017". In the Reference list, data citations must be labeled with "[DATASET]". A data reference must provide the database name, accession number/identifiers and a resolvable link to the landing page from which the data can be accessed at the end of the reference. Further instructions are available at .

<https://www.embopress.org/page/journal/17574684/authorguide#expandedview>

11) For more information: There is space at the end of each article to list relevant web links for further consultation by our readers. Could you identify some relevant ones and provide such information as well? Some examples are patient associations, relevant databases, OMIM/proteins/genes links, author's websites, etc...

12) Author contributions: CRediT has replaced the traditional author contributions section because it offers a systematic machine readable author contributions format that allows for more effective research assessment. Please remove the Authors Contributions from the manuscript and use the free text boxes beneath each contributing author's name in our system to add specific details on the author's contribution. More information is available in our guide to authors.

13) Disclosure statement and competing interests: We updated our journal's competing interests policy in January 2022 and request authors to consider both actual and perceived competing interests. Please review the policy <https://www.embopress.org/competing-interests> and update your competing interests if necessary.

14) Every published paper now includes a 'Synopsis' to further enhance discoverability. Synopses are displayed on the journal webpage and are freely accessible to all readers. They include a short stand first (maximum of 300 characters, including space) as well as 2-5 one-sentences bullet points that summarizes the paper. Please write the bullet points to summarize the key NEW findings. They should be designed to be complementary to the abstract - i.e. not repeat the same text. We encourage inclusion of key acronyms and quantitative information (maximum of 30 words / bullet point). Please use the passive voice. Please attach these in a separate file or send them by email, we will incorporate them accordingly.

Please also suggest a striking image or visual abstract to illustrate your article as a PNG file 550 px wide x 300-600 px high. Share synopsis text and image, as well as eTOC:

Please note that these would be the final versions and changes during proofing are usually not allowed

15) As part of the EMBO Publications transparent editorial process initiative (see our policy here:

https://www.embopress.org/transparent-process#Review_Process), Molecular Systems Biology will publish online a Peer Review File (PRF) to accompany accepted manuscripts.

In the event of acceptance, this file will be published in conjunction with your paper and will include the anonymous referee reports, your point-by-point response and all pertinent correspondence relating to the manuscript. Let us know whether you agree with the publication of the PRF and as here, if you want to remove or not any figures from it prior to publication.

Please note that the Authors checklist will be published at the end of the PRF.

Molecular Systems Biology has a "scooping protection" policy, whereby similar findings that are published by others during review or revision are not a criterion for rejection. Should you decide to submit a revised version, I do ask that you get in touch after three months if you have not completed it, to update us on the status.

I look forward to receiving your revised manuscript.

Yours sincerely,

Poonam Bheda

Poonam Bheda, PhD
Scientific Editor
Molecular Systems Biology

Reviewer #1:

Morgens et al.'s manuscript, "From enhancers to genome conformation: complex transcriptional control underlies expression of a single herpesviral gene," aimed to define the steps of KSHV transcriptional regulation using ORF68 expression as an example. The manuscript is clearly written, and the figures are very well organized.

In the study, the authors generated reporter KSHV with Halo-tag fused to ORF68 and monitored ORF68 protein expression levels. With CRISPRi, a fusion of inactive Cas9 (dCas9) to the Krüppel-associated box (KRAB) repressor along with tiling guide RNAs, the authors comprehensively determined KSHV transcribing regions that are important for ORF68 expression during reactivation in iSLK cells. The study identified the transactivation of six genomic loci positively associated with ORF68 expression, while two were negatively associated with ORF68 expression. Although some of the transcripts (ORF50 and ORF68 itself) whose downregulation impairs the ORF68-reporter signal is not surprising, the authors also identified that ORF75 and one of the non-coding RNA transcription are important for ORF68 expression.

The authors further studied the impact of coding loci on ORF68 expression using Cas9 nuclease and identified that ORF75 protein positively regulates ORF68 expression, which is new to the field. With an isolated reporter study, the authors confirmed that ORF75 alone could activate the ORF68 promoter. Finally, the authors employed Capture Hi-C and revealed that TSS68 and polyA75 are in close proximity to TSS68 and TSS75, respectively, in 3D nuclear space. This study utilized novel techniques and KSHV inducible latent chromosomes as a unique model to shed light on the KSHV transcription program and the association between enhancer-promoter interactions and transcription activation. With the small viral latent chromatin with KSHV reactivation, the authors demonstrated the dynamics of transcription regulation at high resolution. However, additional experiments would be needed to concrete the author's ORF68 transcription regulatory model. The models of TSS68, TSS72, and polyA75 as enhancers for ORF68, in particular, may require additional evidence to justify them. Finally, confirming a result in naturally infected cells would be ideal, as iSLK cells are highly artificial experiment platforms.

Major Comments:

1. In Figure 1, iSLK cells were reactivated by doxycycline and sodium butyrate, and cells were sorted for high Halo-ORF68 and Low Halo-ORF68. However, a low Halo-ORF68 cell population likely contains cells that did not respond to the stimuli, which may compromise the resolution of the analyses. Accordingly, it would be better if the authors show the proportion of ORF68 protein-expressed cells in Fig. 1c. This also applies to the infection rate of KSHV in HEK293 cells shown in Fig 4c. These data are essential to suggest the significance of ORF68 regulation in reactivation.
2. Because ORF75 KO globally downregulates KSHV gene expression (Fig 2A), it is important to clarify whether ORF75 has a direct transactivation role on ORF68 expression or the ORF68 downregulation is an indirect effect (i.e., inhibition of cell signaling events or activation of innate immune responses). Additional experiments such as ChIP-qPCR to show enrichment of ORF75 at promoter region may provide valuable insights into this. Also, it is ideal to complement ORF75 and examine if the ORF75 can rescue ORF68 expression in ORF75 Cas9 cells; these provide more direct evidence for the involvement of ORF75 in ORF68 transcription.
3. In Fig 4B, S6J, the authors present a model showing that TSS72 negatively regulates TSS75, while polyA75 positively regulates TSS75. However, the conclusion is based solely on the negative or positive correlation observed in RNA-seq data. The putative more proximity physical localization than expected linear distance based on contact frequencies is not very strong evidence for the regulatory interaction. Including a functional study to prove if they are indeed transcription regulatory elements is ideal.
4. Similar to the above, the authors state that TSS68, TSS72, and polyA75 function as enhancers because they are located in the insulator region. Additional epigenetic characteristics and transcription-related enzyme tethering should be included/discussed to conclude whether they are enhancers. The correlation of total transcription profiles after sgRNA treatment, as shown in Fig 4a, is not necessarily associated with enhancer function. Similarity of the transcription regulatory profile may suggest the localization within the same topology associate domain that shares the same RNA polymerase II complex localized proximity. Accordingly, the regulation by the polyA75 by the TSS75 described in Fig 4B might be misleading. Please consider.
5. Fig. S6 shows changes in contact frequency between identified loci caused by transcription perturbation with CRISPRi, which is an innovative approach. However, Figure S6H shows that the contact frequency between polyA75 and TSS75 is increased with CRISPRi polyA75, which does not support the model presented in Fig. S6J. Please confirm.
6. Possible indirect effects should be considered. In Fig. 6, it is a little too aggressive to suggest the silencing of specific genomic loci is responsible for the changes in the overall 3D genome structure via enhancer-promoter interaction mechanism during reactivation. KSHV reactivation induces dynamic changes in many viral and host gene expressions, which makes it

unlikely that the silencing of a single gene is responsible for those changes. Fig. 6A may include additional explanations and alternatives.

7. The authors utilized iSLK cells in all their experiments. However, it is important to consider whether similar regulatory mechanisms are applicable to naturally KSHV-infected cells such as BC1 or Bcbl1 cells. A few key experiments should be repeated in naturally infected cells.

Minor comments:

1. Line 108-109: Mention that the percentage of HaloTag-ORF68 cells was measured. However, the figure (Y-axis) demonstrates the relative fraction of cells. Please correct.

2. It would be more accurate to change the label on the Y-axis from 'early gene expression' to 'ORF68 gene expression' in Fig 1B and Fig S1A.

3. Please confirm why the Hi-C data with a 2kb resolution shows more tiles compared to the Hi-C data with a 1kb resolution in Fig 6B and C.

4. Please match the figures and text. The figures are labeled in Capital letters, but figures are mentioned in smaller cases in the text.

Reviewer #2:

The regulation of transcription on compact DNA virus genomes presents unique challenges for regulated expression. While an important and emerging topic there is limited knowledge how this is orchestrated. Here, Morgens et al. take a high-density functional genomics approach to investigate how gene expression is regulated on the genome of the oncogenic DNA virus Kaposi's sarcoma-associated herpesvirus (KSHV). Focusing on the regulation of ORF68, a gene that encodes a critical protein in the KSHV lifecycle, the authors establish a clever high-density CRISPRi and CRISPR knockout approach and identify viral loci that facilitate ORF68 expression. Interestingly, this approach discovers both coding and noncoding distal regions of the viral genome are important. Capture Hi-C further facilitated the mechanistic dissection of ORF68 regulation, including the identification of an insulated neighborhood near the ORF68 locus critical for its proper regulation. The experimental approach and analyses are well described and thorough. The findings are important as they describe a detailed mechanism of gene regulation through both coding and noncoding elements on a compact DNA viral genome. Overall, this is a strong manuscript and I have limited comments.

One aspect of gene regulation, and in particular genome organization, that is lacking, is the epigenetic status of the identified loci. A number of studies have published genome-wide epigenetic data on the KSHV genome or have been performed in KSHV-infected cells. Intersection analyses of ORF68 regulatory loci with these data and commenting on their status may help to clarify whether there are any unique epigenetic signatures present.

Reviewer #3:

In this manuscript, Morgens and colleagues have done impressive work dissecting the complex regulatory network controlling the expression of ORF68, a critical viral gene for KSHV infection. The authors' use of several different approaches, including CRISPRi screening and capture Hi-C, has provided robust evidence regarding the role of 3D KSHV genome conformation in temporally regulating viral gene expression during the KSHV lifecycle. The results presented are compelling and mainly support the authors' conclusion. However, a few questions directly spun from the results, and that, if addressed, will further strengthen the significance of this work. For example, the data from CRISPRi suggested that genetic manipulation at specific loci of KSHV alters ORF68 expression by changing the 3D structure of the viral genome. Since several groups reported that CTCF and Cohesin complex bind to the KSHV genome and CTCF/Cohesin interaction is essential in regulating chromatin loop formation, does CRISPRi manipulation alter CTCF and Cohesin binding at the identified regulatory regions? Addressing these questions could potentially increase our understanding of KSHV infection.

Moreover, the authors suggested that these regulatory regions may function as viral enhancers. If so, are these regions enriched with H3K27ac and H3K4me1 enhancer-specific markers? Do these markers change during CRISPRi perturbation and lytic reactivations, which are known to be associated with changes in viral gene expression? Ideally, H3K27ac Hi-ChIP would be a strong complementing approach to further support the authors' conclusion.

Finally, there is a minor observation: at lane 286, the authors wrote, "...and the irregular shape of the loops we do see." It is not

entirely clear to this reviewer what the authors mean or how the (irregular) shape of the loops can be inferred from Figure 5d. A clarification would enhance the clarity of the text. For instance, are the loops in Figure 5d not uniformly circular or do they exhibit other irregularities in their shape?

Response to reviewer comments

Reviewer #1:

Morgens et al.'s manuscript, "From enhancers to genome conformation: complex transcriptional control underlies expression of a single herpesviral gene," aimed to define the steps of KSHV transcriptional regulation using ORF68 expression as an example. The manuscript is clearly written, and the figures are very well organized.

In the study, the authors generated reporter KSHV with Halo-tag fused to ORF68 and monitored ORF68 protein expression levels. With CRISPRi, a fusion of inactive Cas9 (dCas9) to the Krüppel-associated box (KRAB) repressor along with tiling guide RNAs, the authors comprehensively determined KSHV transcribing regions that are important for ORF68 expression during reactivation in iSLK cells. The study identified the transactivation of six genomic loci positively associated with ORF68 expression, while two were negatively associated with ORF68 expression. Although some of the transcripts (ORF50 and ORF68 itself) whose downregulation impairs the ORF68-reporter signal is not surprising, the authors also identified that ORF75 and one of the non-coding RNA transcription are important for ORF68 expression.

The authors further studied the impact of coding loci on ORF68 expression using Cas9 nuclease and identified that ORF75 protein positively regulates ORF68 expression, which is new to the field. With an isolated reporter study, the authors confirmed that ORF75 alone could activate the ORF68 promoter. Finally, the authors employed Capture Hi-C and revealed that TSSalt and polyA75 are in close proximity to TSS68 and TSS75, respectively, in 3D nuclear space. This study utilized novel techniques and KSHV inducible latent chromosomes as a unique model to shed light on the KSHV transcription program and the association between enhancer-promoter interactions and transcription activation. With the small viral latent chromatin with KSHV reactivation, the authors demonstrated the dynamics of transcription regulation at high resolution. However, additional experiments would be needed to concrete the author's ORF68 transcription regulatory model. The models of TSSalt, TSS72, and polyA75 as enhancers for ORF68, in particular, may require additional evidence to justify them. Finally, confirming a result in naturally infected cells would be ideal, as iSLK cells are highly artificial experiment platforms.

Major Comments:

1. In Figure 1, iSLK cells were reactivated by doxycycline and sodium butyrate, and cells were sorted for high Halo-ORF68 and Low Halo-ORF68. However, a low Halo-ORF68 cell population likely contains cells that did not respond to the stimuli, which may compromise the resolution of the analyses. Accordingly, it would be better if the authors show the proportion of ORF68 protein-expressed cells in Fig. 1c. This also applies to the infection rate of KSHV in HEK293 cells shown in Fig 4c. These data are essential to suggest the significance of ORF68 regulation in reactivation.

We thank the reviewer for the suggestion to increase interpretability of the results we present. We have replaced the normalized data with the raw proportion of ORF68 expressing cells in Figure 1c and the unnormalized infection rates in Figure 4c as suggested.

2. Because ORF75 KO globally downregulates KSHV gene expression (Fig 2A), it is important to clarify whether ORF75 has a direct transactivation role on ORF68 expression or the ORF68 downregulation is an indirect effect (i.e., inhibition of cell signaling events or activation of innate immune responses). Additional experiments such as ChIP-qPCR to show enrichment of ORF75 at promoter region may provide valuable insights into this. Also, it is ideal to complement ORF75 and examine if the ORF75 can rescue ORF68 expression in ORF75 Cas9 cells; these provide more direct evidence for the involvement of ORF75 in ORF68 transcription.

Our data shows that ORF75 is functionally required for ORF68 expression using independent targeting of either the promoter or the coding region, and that ORF75 is sufficient for activating an ORF68 promoter in HEK293T cells. We do not have any additional data suggesting whether ORF75 is acting as a DNA-binding factor, now further clarified in the text. However, based on previous data, we believe it is most likely that ORF75 is acting to antagonize host factors which may more directly antagonize viral gene expression (Full et al 2014 *PLoS Pathogens*). Indeed, that previous study demonstrated a partial rescue of a ORF75 stop virus from ORF75 overexpression. Although we attempted this experiment as the reviewer requested, we consistently encountered the technical problem of ORF75 transduction in iSLKs significantly decreasing reactivation efficiency in wildtype cells, and thus we were not able to clearly interpret the results.

3. In Fig 4B, S6J, the authors present a model showing that TSS72 negatively regulates TSS75, while polyA75 positively regulates TSS75. However, the conclusion is based solely on the negative or positive correlation observed in RNA-seq data. The putative more proximity physical localization than expected linear distance based on contact frequencies is not very strong evidence for the regulatory interaction. Including a functional study to prove if they are indeed transcription regulatory elements is ideal.

Based on our data on RNA levels (Figure 2a, Figure EV2) and nascent RNA levels (Figure 2e), we are confident that TSS72 and polyA75 do regulate ORF75 RNA expression. However, we have now clarified in the text that we do not know if this is happening as a cis regulatory element

Our Hi-C data is consistent with a traditional regulatory element model where the element and the promoter are in close physical proximity to each other but does not rule out other models. Our Cas9 data (Figure 3) show that TSS72 and polyA75 are not acting through mechanisms that could be disrupted by Cas9 nuclease (i.e. proteins), but this does not rule out alternative mechanisms. We now present new data per reviewer request on the chromatin state of these loci throughout the course of early reactivation and show that TSS72 (but not polyA75) is located within an H3K27ac positive peak, consistent with an enhancer-like activity.

4. Similar to the above, the authors state that TSSalt, TSS72, and polyA75 function as enhancers because they are located in the insulator region. Additional epigenetic characteristics and transcription-related enzyme tethering should be included/discussed to conclude whether they are enhancers. The correlation of total transcription profiles after sgRNA treatment, as shown in Fig 4a, is not necessarily associated with enhancer function. Similarity of the transcription regulatory profile may suggest the localization within the same topology associate domain that shares the same RNA polymerase II complex localized proximity. Accordingly, the regulation by the polyA75 by the TSS75 described in Fig 4B might be misleading. Please consider.

Thank you to the reviewer for their suggestion; we now included H3K27ac, H3K4me1, and CTCF data throughout the early reactivation to assess the states of TSSalt, TSS72, and polyA75. We identified that TSSalt and TSS72 are located within a H3K27ac peak, but polyA75 is not. And while polyA75 has H3K4me1 marks, the other two do not. CTCF binding is also found at TSSalt and TSS72. These observations are consistent with TSSalt and TSS72 acting as enhancers but does not rule out alternate explanations. As for whether, for example, the connection between polyA75 and TSS75 could be explained by shared regulatory domains, we include discussion of this possibility, though we can say that targeting the gene body of ORF75 (Between TSS75 and polyA75) does not have the same effect.

5. Fig. S6 shows changes in contact frequency between identified loci caused by transcription perturbation with CRISPRi, which is an innovative approach. However, Figure S6H shows that the contact frequency between polyA75 and TSS75 is increased with CRISPRi polyA75, which does not support the model presented in Fig. S6J. Please confirm.

In figure EV5H, we observe that when targeting TSS75, the contact frequency between TSS75 and polyA75 is unchanged relative to the unreacted control. Similarly, targeting polyA75 also leaves that interaction unchanged. Indeed, we see that in Figure EV5D, targeting TSS75 or polyA75 has a very similar effect on the globally 3D structure of the genome, consistent with the model presented in EV5J that polyA75 has a positive regulatory effect on TSS75.

Our model for this result, which we now clarify in text, is that ORF75 protein expression is necessary for the large rearrangement we observe in the 3D structure of that region of the viral genome. That change results in polyA75 and TSS75 having a reduced contact frequency (perhaps due to diluting their interactions across the now larger regulatory domain).

6. Possible indirect effects should be considered. In Fig. 6, it is a little too aggressive to suggest the silencing of specific genomic loci is responsible for the changes in the overall 3D genome structure via enhancer-promoter interaction mechanism during reactivation. KSHV reactivation induces dynamic changes in many viral and host gene expressions, which makes it unlikely that the silencing of a single gene is responsible for those changes. Fig. 6A may include additional explanations and alternatives.

We have clarified our claims in text but based on our observation in Figure EV6D-F, it is likely that most if not all of the changes to the 3D structure are mediated directly or indirectly by the expression of viral proteins, not on the changes to the regulatory elements.

7. The authors utilized iSLK cells in all their experiments. However, it is important to consider whether similar regulatory mechanisms are applicable to naturally KSHV-infected cells such as BC1 or Bcb1 cells. A few key experiments should be repeated in naturally infected cells.

Thank you for the suggestion. iSLK cells are a workable and manipulable model for KSHV infections with a high efficiency of reactivation and a low viral copy number, making them ideal for many of the large-scale experiments performed here. They also represent a different class of cell type (epithelial) than natural tropisms of B-cells or endothelial cells. It is of key interest to us in future work to consider how these regulatory networks are conserved or not between cell types. Our expectation is that if these regulatory regions are acting as enhancers, then their activity would be mediated by host transcription factor binding, which itself varies widely between cell types. Thus, we might expect the viral protein players to be relatively conserved, but the noncoding regulators to be variable. Indeed, as described below, this is largely what we observed.

We acquired a CRISPRi+ KSHV-infected B cell line (BCBL1) to test whether we observe the same regulatory regions affecting ORF68 expression or viral production and have now included that data (Figure EV2e; EV4b). In short, we observe that the promoter regions driving ORF50, 68, and 75 also contribute to ORF68 expression and are essential for infectious virion production in BCBL1 cells, but many of the non-coding elements identified in iSLK cells (with the exception of TSS57) do not contribute to ORF68 levels in BCBL1 cells.

As described above, this is consistent with our expectation. However, there are several technical limitations which prevent us from concluding that this represents a cell-type specific regulatory network. 1) Not only noncoding elements, but transcriptional start sites and chromatin states vary between cell types, meaning the same guides that are effective in iSLKs may not be in BCBL1s. 2) A direct comparison is difficult but ORF75 RNA expression appears much lower in BCBLs, with only a ~5 fold increase upon reactivation (vs a ~100 fold increase in iSLKs). This may suggest that iSLKs are more heavily dependent on this ORF75-linked regulation than BCBL1s. 3) The effectiveness of CRISPRi may be lower in BCBL1s due to the 100s of viral copies present (or the different reactivation conditions), limiting our ability to measure smaller effects. Without the ability to install reporters into the viral genome in these B cells, addressing these challenges in patient-derived lines remains an exciting avenue of research for the future.

Minor comments:

1. Line 108-109: Mention that the percentage of HaloTag-ORF68 cells was measured. However, the figure (Y-axis) demonstrates the relative fraction of cells. Please correct.
2. It would be more accurate to change the label on the Y-axis from 'early gene expression' to 'ORF68 gene expression' in Fig 1B and Fig S1A.
3. Please confirm why the Hi-C data with a 2kb resolution shows more tiles compared to the Hi-C data with a 1kb resolution in Fig 6B and C.
4. Please match the figures and text. The figures are labeled in Capital letters, but figures are mentioned in smaller cases in the text.

Thank you for these comments; they have all been fixed. For the HiC data in Figures 6B, C, and D, these are only showing the tiles from the 100-152kb region of the viral genome. We have added additional clarification to the figure legends.

Reviewer #2:

The regulation of transcription on compact DNA virus genomes presents unique challenges for regulated expression. While an important and emerging topic there is limited knowledge how this is orchestrated. Here, Morgens et al. take a high-density functional genomics approach to investigate how gene expression is regulated on the genome of the oncogenic DNA virus Kaposi's sarcoma-associated herpesvirus (KSHV). Focusing on the regulation of ORF68, a gene that encodes a critical protein in the KSHV lifecycle, the authors establish a clever high-density CRISPRi and CRISPR knockout approach and identify viral loci that facilitate ORF68 expression. Interestingly, this approach discovers both coding and noncoding distal regions of the viral genome are important. Capture Hi-C further facilitated the mechanistic dissection of ORF68 regulation, including the identification of an insulated neighborhood near the ORF68 locus critical for its proper regulation. The experimental approach and analyses are well described and thorough. The findings are important as they describe a detailed mechanism of gene regulation through both coding and noncoding elements on a compact DNA viral genome. Overall, this is a strong manuscript and I have limited comments.

One aspect of gene regulation, and in particular genome organization, that is lacking, is the epigenetic status of the identified loci. A number of studies have published genome-wide epigenetic data on the KSHV genome or have been performed in KSHV-infected cells. Intersection analyses of ORF68 regulatory loci with these data and commenting on their status may help to clarify whether there are any unique epigenetic signatures present.

Thank you for the reviewer's positive and supportive comments. Per suggestion, we have included CUT&RUN data for H3K27ac, H3K4me1, and CTCF across the viral genome through the stages of early reactivation. We identified that TSSalt and TSS72 were H3K27ac+, H3K4me1-, CTCF+, a signature consistent with enhancer activity but not uniquely present at these loci. Intriguingly, this signature is also found at the promoter of ORF75 with unknown significance. PolyA75, consistent with a lack of nearby TSS, was instead H3K27ac-, H3K4me1+, CTCF-, perhaps indicating it acts as an enhancer in some untested condition or that it acts through an unknown mechanism independent of these marks.

Reviewer #3:

In this manuscript, Morgens and colleagues have done impressive work dissecting the complex regulatory network controlling the expression of ORF68, a critical viral gene for KSHV infection. The authors' use of several different approaches, including CRISPRi screening and capture Hi-C, has provided robust evidence regarding the role of 3D KSHV genome conformation in temporally regulating viral gene expression during the KSHV lifecycle. The results presented are compelling and mainly support the authors' conclusion. However, a few questions directly spun from the results, and that, if addressed, will further strengthen the significance of this work. For example, the data from CRISPRi suggested that genetic manipulation at specific loci of KSHV alters ORF68 expression by changing the 3D structure of the viral genome. Since several groups reported that CTCF and Cohesin complex bind to the KSHV genome and CTCF/Cohesin interaction is essential in regulating chromatin loop formation, does CRISPRi manipulation alter CTCF and Cohesin binding at the identified regulatory regions? Addressing these questions could potentially increase our understanding of KSHV infection.

Thank you very much for the positive and supportive remarks! To directly test their question on CTCF regulation of the 3D structure, we performed CTCF CUT&RUN while perturbing the key viral proteins ORF50 and ORF75. We observed very few dynamic peaks, with most CTCF signal being highly stable except for the appearance of a small peak at the TSS72 region. This stability in CTCF binding while we observe many changes to the 3D structure may indicate that this is mediated by another factor such as cohesion or YY1.

Moreover, the authors suggested that these regulatory regions may function as viral enhancers. If so, are these regions enriched with H3K27ac and H3K4me1 enhancer-specific markers? Do these markers change during CRISPRi perturbation and lytic reactivations, which are known to be associated with changes in viral gene expression? Ideally, H3K27ac Hi-ChIP would be a strong complementing approach to further support the authors' conclusion.

Per the reviewer's suggestion, we also performed H3K27ac and H3K4me1 CUT&RUN, and while global levels of these marks on the viral genome are dramatically altered upon reactivation, we do not see other changes upon knockout of ORF50 or ORF75. Two of the regulatory elements identified, TSS72 and TSSalt, do maintain H3K27ac marks (and CTCF binding), consistent with active enhancer activity.

Finally, there is a minor observation: at lane 286, the authors wrote, "...and the irregular shape of the loops we do see." It is not entirely clear to this reviewer what the authors mean or how the (irregular) shape of the loops can be inferred from Figure 5d. A clarification would enhance the clarity of the text. For instance, are the loops in Figure 5d not uniformly circular or do they exhibit other irregularities in their shape?

Thank you for the comments. We have now clarified this remark in text. In short, the signatures of loops we observe in Figure 5d, are not uniformly circular, but instead appear rectangular, for example with a 14 kb region strongly interacting with a 6 kb region.

24th Oct 2024

Manuscript Number: MSB-2024-12411R

Title: From enhancers to genome conformation: complex transcriptional control of a single herpesviral gene

Dear Dr. Glaunsinger,

Thank you for the submission of your revised manuscript to Molecular Systems Biology. We have now received the enclosed reports from the referees that were asked to re-assess it. As you will see the reviewers are now globally supportive and I am pleased to inform you that we will be able to accept your manuscript pending the following final amendments:

1) The title should refrain from using punctuation. Please edit accordingly.

2) Please include keywords to max. 5.

3) Please format the Data availability section according to the example below:

"The datasets and computer code produced in this study are available in the following databases:

- Chip-Seq data: Gene Expression Omnibus GSE46748 (<https://www.ncbi.nlm.nih.gov/geo/query/acc.cgi?acc=GSE46748>)

- Modeling computer scripts: GitHub (<https://github.com/SysBioChalmers/GECKO/releases/tag/v1.0>)

- [data type]: [full name of the resource] [accession number/identifier] ([doi or URL or identifiers.org/DATABASE:ACCESSION])"

4) Please rename "Conflict of Interest" to "Disclosure and competing interests statement". We updated our journal's competing interests policy in January 2022 and request authors to consider both actual and perceived competing interests. Please review the policy <https://www.embopress.org/competing-interests> and update your competing interests if necessary.

5) References: Please correct the reference citation in the reference list - references should be given alphabetically, not numbered. Where there are more than 10 authors on a paper, please only list the first 10, followed by "et al.". Please check "Author Guidelines" for more information.

<https://www.embopress.org/page/journal/17574684/authorguide#referencesformat>

6) All Materials and Methods need to be described in the main text using our 'Structured Methods' format. According to this format, the Methods section includes a Reagents and Tools Table (listing key reagents, experimental models, software and relevant equipment and including their sources and relevant identifiers) followed by a Methods and Protocols section describing the methods, ideally using a step-by-step protocol format. The aim is to facilitate adoption of the methodologies across labs. Please download and fill our Reagents and Tools Table template (.docx), which you can find in our author guidelines:

7) In the Methods, please take care of the following:

- Cell lines: Please include all information requested in the author checklist for cell lines used in the manuscript - currently catalog numbers are missing. This information should be included in the Reagents and Tools table. Please also be sure to include a sentence in the Methods as to whether or not the cell lines were tested for mycoplasma contamination.

8) Please place individual sections of the manuscript in the following order: Title page - Abstract & Keywords - Introduction - Results - Discussion - Methods - Data Availability - Acknowledgements - Disclosure and Competing Interests Statement - References - Figure Legends - Expanded View Figure Legends.

9) For the figures and figure legends, please take care of the following:

- Exact p value are missing for figure EV2 D. This may be provided in the figure or figure legend.

- Please note that information related to n is missing in the legend of figure 6H-J.

- Please note that the measure of center for the error bars needs to be defined in the legends of figures 1C; 2E; 3I, J; 4C; EV2 C-E; EV3 B.

10) Dataset EV legends: Supplementary Data with Dataset labels should be removed from the manuscript file; Dataset legends should be uploaded as a separate tab in each Excel file (there descriptions in some, but they should be labeled as Dataset EVx).

11) Appendix file: Please upload the Appendix as a single PDF (no separate image files are needed).

12) Funding: Please ensure that all funding sources are entered into the manuscript submission system. Currently NIH S10 OD018174 Instrumentation Grant is missing in our system.

13) As part of the EMBO Publications transparent editorial process initiative (see our policy here:

https://www.embopress.org/transparent-process#Review_Process), Molecular Systems Biology will publish online a Peer Review File (PRF) to accompany accepted manuscripts. This file will be published in conjunction with your paper and will include the anonymous referee reports, your point-by-point response and all pertinent correspondence relating to the manuscript. Let us know whether you agree with the publication of the PRF and as here, if you want to remove or not any figures from it prior to publication. Please note that the Authors checklist will be published at the end of the PRF.

14) Please provide a point-by-point letter INCLUDING my comments and your detailed responses (as Word file).

I look forward to reading a new revised version of your manuscript as soon as possible.

Yours sincerely,

Poonam Bheda, PhD
Scientific Editor
Molecular Systems Biology

Reviewer #1:

The authors responded to previous critiques very well-no further concerns from this reviewer.

Reviewer #3:

In this updated version of the manuscript, the authors included additional results that effectively address the previous critiques, thereby strengthening their conclusions and the significance of the data.

24th Oct 2024

Manuscript Number: MSB-2024-12411R

Title: From enhancers to genome conformation: complex transcriptional control of a single herpesviral gene

Dear Dr. Glaunsinger,

Thank you for the submission of your revised manuscript to Molecular Systems Biology. We have now received the enclosed reports from the referees that were asked to re-assess it. As you will see the reviewers are now globally supportive and I am pleased to inform you that we will be able to accept your manuscript pending the following final amendments:

1) The title should refrain from using punctuation. Please edit accordingly.

2) Please include keywords to max. 5.

3) Please format the Data availability section according to the example below:

"The datasets and computer code produced in this study are available in the following databases:

- Chip-Seq data: Gene Expression Omnibus GSE46748

(<https://www.ncbi.nlm.nih.gov/geo/query/acc.cgi?acc=GSE46748>)

- Modeling computer scripts: GitHub (<https://github.com/SysBioChalmers/GECKO/releases/tag/v1.0>)

- [data type]: [full name of the resource] [accession number/identifier] ([doi or URL

or identifiers.org/DATABASE:ACCESSION)"

4) Please rename "Conflict of Interest" to "Disclosure and competing interests statement". We updated our journal's competing interests policy in January 2022 and request authors to consider both actual and perceived competing interests. Please review the policy <https://www.embopress.org/competing-interests> and update your competing interests if necessary.

5) References: Please correct the reference citation in the reference list - references should be given alphabetically, not numbered. Where there are more than 10 authors on a paper, please only list the first 10, followed by "et al.". Please check "Author Guidelines" for more information.

<https://www.embopress.org/page/journal/17574684/authorguide#referencesformat>

6) All Materials and Methods need to be described in the main text using our 'Structured Methods' format. According to this format, the Methods section includes a Reagents and Tools Table (listing key reagents, experimental models, software and relevant equipment and including their sources and relevant identifiers) followed by a Methods and Protocols section describing the methods, ideally using a step-by-step protocol format. The aim is to facilitate adoption of the methodologies across labs.

Please download and fill our Reagents and Tools Table template (.docx), which you can find in our author guidelines: <https://www.embopress.org/page/journal/14693178/authorguide#structuredmethods>.

An example of a Method paper with Structured Methods can be found

here: <https://www.embopress.org/doi/10.15252/msb.20178071>.

7) In the Methods, please take care of the following:

- Cell lines: Please include all information requested in the author checklist for cell lines used in the manuscript - currently catalog numbers are missing. This information should be included in the Reagents and Tools table. Please also be sure to include a sentence in the Methods as to whether or not the cell lines were tested for mycoplasma contamination.

8) Please place individual sections of the manuscript in the following order: Title page - Abstract & Keywords - Introduction - Results - Discussion - Methods - Data Availability - Acknowledgements - Disclosure and Competing Interests Statement - References - Figure Legends - Expanded View Figure Legends.

9) For the figures and figure legends, please take care of the following:

- Exact p value are missing for figure EV2 D. This may be provided in the figure or figure legend.

- Please note that information related to n is missing in the legend of figure 6H-J.

- Please note that the measure of center for the error bars needs to be defined in the legends of figures 1C; 2E; 3I, J; 4C; EV2 C-E; EV3 B.
- 10) Dataset EV legends: Supplementary Data with Dataset labels should be removed from the manuscript file; Dataset legends should be uploaded as a separate tab in each Excel file (there descriptions in some, but they should be labeled as Dataset EVx).
- 11) Appendix file: Please upload the Appendix as a single PDF (no separate image files are needed).
- 12) Funding: Please ensure that all funding sources are entered into the manuscript submission system. Currently NIH S10 OD018174 Instrumentation Grant is missing in our system.
- 13) As part of the EMBO Publications transparent editorial process initiative (see our policy here: https://www.embopress.org/transparent-process#Review_Process), Molecular Systems Biology will publish online a Peer Review File (PRF) to accompany accepted manuscripts. This file will be published in conjunction with your paper and will include the anonymous referee reports, your point-by-point response and all pertinent correspondence relating to the manuscript. Let us know whether you agree with the publication of the PRF and as here, if you want to remove or not any figures from it prior to publication. Please note that the Authors checklist will be published at the end of the PRF.
- 14) Please provide a point-by-point letter INCLUDING my comments and your detailed responses (as Word file).

I look forward to reading a new revised version of your manuscript as soon as possible.

Yours sincerely,

Poonam Bheda, PhD
Scientific Editor
Molecular Systems Biology

Thank you for the positive comments. We have addressed the editorial comments and highlighted the changes in the manuscript text: 1) A new title has been included. 2) Keywords have been included. 3) Data availability has been reformatted. 4) Conflict of interest statement has been renamed and rechecked for journal policy. 5) Reference have been reformatted to match EMBO press guidelines. 6) A reagents table is now provided. 7) Additional cell line info has been provided. 8) Manuscript text has been reordered. 9) Missing information has been added to indicated figure legends. 10) Dataset labels have been removed, and datasets have been updated to include a separate tab with the appropriate Dataset label. 11) No appendix is provided. 12) The instrumentation grant was included in error and has been removed from the acknowledgements. 13) The authors agree with the publication of the PRF. 14) We have provided this point-by-point response, which includes editorial communication and the previous point-by-point response.

Reviewer #1:

The authors responded to previous critiques very well-no further concerns from this reviewer.

Reviewer #3:

In this updated version of the manuscript, the authors included additional results that effectively address the previous critiques, thereby strengthening their conclusions and the significance of the data.

31st Oct 2024

Manuscript number: MSB-2024-12411RR

Title: Enhancers and genome conformation provide complex transcriptional control of a herpesviral gene

Dear Dr. Glaunsinger,

Congratulations on an excellent manuscript, I am pleased to inform you that your manuscript has been accepted for publication in Molecular Systems Biology. Thank you for your comprehensive response to referee concerns. It has been a pleasure to work with you to get this to the acceptance stage.

Yours sincerely,

Poonam Bheda, PhD
Scientific Editor
Molecular Systems Biology
